# Data Models for Dataset Drift Controls in Machine Learning With Optical Images

**Luis Oala**[*]                                     *luis.oala@dotphoton.com*
*Fraunhofer HHI and Dotphoton AG*

**Marco Aversa**[*]                                  *marco.aversa@dotphoton.com*
*Dotphoton AG and University of Glasgow*

**Gabriel Nobis**                                    *gabriel.nobis@hhi.fraunhofer.de*
*Fraunhofer HHI*

**Kurt Willis**                                      *kurt.willis@hhi.fraunhofer.de*
*Fraunhofer HHI*

**Yoan Neuenschwander**                              *yoan.neuenschwander@hesge.ch*
*HEPIA/HES-SO*

**Michèle Buck**                                     *michele.kyncl@tum.de*
*Klinikum rechts der Isar*

**Christian Matek**                                  *christian.matek@helmholtz-muenchen.de*
*Helmholtz Zentrum Munich*

**Jérôme Extermann**                                 *jerome.extermann@hesge.ch*
*HEPIA/HES-SO*

**Enrico Pomarico**                                  *enrico.pomarico@hesge.ch*
*HEPIA/HES-SO*

**Wojciech Samek**                                   *wojciech.samek@hhi.fraunhofer.de*
*Fraunhofer HHI*

**Roderick Murray-Smith**                            *roderick.murray-smith@glasgow.ac.uk*
*University of Glasgow*

**Christoph Clausen**                                *christoph.clausen@dotphoton.com*
*Dotphoton AG*

**Bruno Sanguinetti**                                *bruno.sanguinetti@dotphoton.com*
*Dotphoton AG*

**Reviewed on OpenReview:** *https://openreview.net/forum?id=I4IkGmgFJz*

---

[*]Equal contribution

## Abstract

Camera images are ubiquitous in machine learning research. They also play a central role in the delivery of important public services spanning medicine or environmental surveying. However, the application of machine learning models in these domains has been limited because of robustness concerns. A primary failure mode are performance drops due to differences between the training and deployment data. While there are methods to prospectively validate the robustness of machine learning models to such dataset drifts, existing approaches do not account for explicit models of machine learning's primary object of interest: the data. This limits our ability to study and understand the relationship between data generation and downstream machine learning model performance in a physically accurate manner. In this study, we demonstrate how to overcome this limitation by pairing traditional machine learning with physical optics to obtain explicit and differentiable data models. We demonstrate how such data models can be constructed for image data and used to control downstream machine learning model performance related to dataset drift. The findings are distilled into three applications. First, drift synthesis enables the controlled generation of physically faithful drift test cases to power model selection and targeted generalization. Second, the gradient connection between machine learning task model and data model allows advanced, precise tolerancing of task model sensitivity to changes in the data generation. These drift forensics can be used to precisely specify the acceptable data environments in which a task model may be run. Third, drift optimization opens up the possibility to create drifts that can help the task model learn better faster, effectively optimizing the data generating process itself to support the downstream machine vision task. This is an interesting upgrade to existing imaging pipelines which traditionally have been optimized to be consumed by human users but not machine learning models. The data models require access to raw sensor images as commonly processed at scale in industry domains such as microscopy, biomedicine, autonomous vehicles or remote sensing. Alongside the data model code we release two datasets to the public that we collected as part of this work. In total, the two datasets, Raw-Microscopy and Raw-Drone, comprise 1,488 scientifically calibrated reference raw sensor measurements, 8,928 raw intensity variations as well as 17,856 images processed through twelve data models with different configurations. A guide to access the open code and datasets is available at `https://github.com/aiaudit-org/raw2logit`.

## 1 Introduction

In this study we demonstrate how explicit data models for images can be constructed and then be used for advanced dataset drift controls in machine learning workflows. We connect raw image data, differentiable data models and the standard machine learning pipeline. This combination enables novel, physically faithful validation protocols that can be used towards intended use specifications of machine learning systems, a necessary pre-requisite for the use of any technology in many application domains such as medicine or autonomous vehicles.

Camera image data are a staple of machine learning research, from the early proliferation of neural networks on MNIST [1–4] to leaps in deep learning on CIFAR and ImageNet [5–7] or high-dimensional generative models [8, 9]. Camera images also play an important role in the delivery of various high-impact public and commercial services. Unsurprisingly, the exceptional capacity of deep supervised learning has inspired great imagination to automate or enhance such services. During the 2010s, "deep learning for ..." rang loud in most application domains under the sun, and beyond [10], spanning medicine and biology (microscopy for cell detection [11–14], histopathology [15, 16], opthalmology [17–19], malaria detection [20–23]), geospatial modelling (climate [24–26], precision farming [27–29], pollution detection [30–32]) and more.

However, the excitement was reined in by calls for caution. Machine learning systems exhibit particular failure modes that are contingent on the makeup of their inputs [33–35]. Many findings from the machine learning robustness literature confirm supervised learning's tremendous capacity for identifying features in the training

inputs that are correlated with the true labels [36–40]. But these findings also point to a flipside of this capacity: the sensitivity of the resulting machine learning model's performance to changes - both large and small - in the input data. Because this dependency touches on generalization, a *summum bonum* of machine learning, the implications have been studied across most of its many sub-disciplines including robustness validation [41–54], formal model verification [55–71], uncertainty quantification [72–82], out-of-distribution detection [34, 83–87], semi- [88–90] and self-supervised learning [91, 92], learning theory and optimization [93–96], federated learning [97–99], or compression [100–102], among others.

We refer to the mechanism underlying changes in the input data as dataset drift[1]. Formally, we characterize it as follows. Let $(\boldsymbol{X}_{RAW}, Y) : \Omega \to \mathbb{R}^{H,W} \times \mathcal{Y}$ be the raw sensor data generating random variable[2] on some probability space $(\Omega, \mathcal{F}, \mathbb{P})$, for example with $\mathcal{Y} = \{0, 1\}^K$ for a classification task. Raw inputs $\boldsymbol{x}_{\mathrm{RAW}}$ are in a data state before further processing is applied, in our case photons captured by the pixels of a camera sensor as displayed in the outputs of the "Measurement" block in Figure 1. The raw inputs $\boldsymbol{x}_{\mathrm{RAW}}$ are then further processed by a *data model* $\boldsymbol{\Phi}_{\mathrm{Proc}} : \mathbb{R}^{H,W} \to \mathbb{R}^{C,H,W}$, in our case the measurement hardware like a camera itself or other downstream data processing pipelines, to produce a processed view $\boldsymbol{v} = \boldsymbol{\Phi}_{\mathrm{Proc}}(\boldsymbol{x}_{\mathrm{RAW}})$ of the data as illustrated in the output of the "Data model" block in Figure 1. This processed view $\boldsymbol{v}$ could for example be the finished RGB image, the image data state that most machine learning researchers typically work with to train a *task model* $\boldsymbol{\Phi}_{\mathrm{Task}} : \mathbb{R}^{C,H,W} \to \mathcal{Y}$. Thus, in the conventional machine learning setting we obtain $\boldsymbol{V} = \boldsymbol{\Phi}_{\mathrm{Proc}}(\boldsymbol{X}_{RAW})$ as the image data generating random variable with the target distribution $\mathcal{D}_t = \mathbb{P} \circ (\boldsymbol{V}, Y)^{-1}$. A different data model $\tilde{\boldsymbol{\Phi}}_{\mathrm{Proc}}$ generates a different view $\tilde{\boldsymbol{V}} = \tilde{\boldsymbol{\Phi}}_{\mathrm{Proc}}(\boldsymbol{X}_{RAW})$ of the same underlying raw sensor data generating random variable $\boldsymbol{X}_{RAW}$, resulting in the *dataset drift*

$$\mathcal{D}_s = \mathbb{P} \circ (\tilde{\boldsymbol{V}}, Y)^{-1} \neq \mathcal{D}_t. \tag{1}$$

This characterization of dataset drift is closely related to the concept of distributional robustness in the sense of Huber where "the shape of the true underlying distribution deviates slightly from the assumed model" [104]. In the distributional robustness literature it is typically assumed that *something* changes from $\mathcal{D}_s$ to $\mathcal{D}_t$, however without exactly specifying *what* changes. This makes it unclear what is actually being compared, whether the considered change is plausible and what factors contribute to the change. Data models offer a way out of this ambiguity through exact description of the data generating process. In practice, a common reason for dataset drift in images is a change in the camera types or settings, for example different acquisition microscopes across different lab sites $\boldsymbol{s}$ and $\boldsymbol{t}$ that lead to drifted distributions $\mathcal{D}_s \neq \mathcal{D}_t$. Anticipating and validating the robustness of a machine learning model to these variations in an exact and traceable way is not just an engineering concern but also mandated by quality standards in many industries [105–107]. Omissions to perform physically faithful robustness validations has, among other reasons, slowed or prevented the rollout of machine learning technology in impactful applications such as large-scale automated retinopathy screening [108], machine learning melanoma detection [109, 110] or yield prediction [111] from drone cameras.

Hence, the calls for realistic robustness validation of image machine learning systems are not merely an exercise in intellectual novelty but a matter of integrating machine learning research with real world infrastructures and performance expectations around its core ingredient: the data.

## 1.1 The status quo of dataset drift controls for images

How can one go about validating a machine learning model's performance under image dataset drift? The dominant empirical techniques can broadly be categorized into augmentation and catalogue testing. Each approach has particular benefits and limitations (see Table 1 for a conceptual comparison).

Augmentation testing involves the application of perturbations, for example Gaussian noise, to already processed images [43, 112, 113] in order to approximate the effect of dataset drift. Given a processed dataset

---

[1]Note that the nomenclature around dataset drift is as heterogenous as the disciplines in which it is studied. See [103] for a good discussion of cross-disciplinary taxonomy. Here we are concerned with dataset drift as defined in Equation (1), that is changes in $V$ that are induced by changes in $\boldsymbol{\Phi}_{\mathrm{Proc}}$ which some works also refer to as covariate shift or more generally as distribution shift.

[2]We write an uppercase letter $A$ for a real valued random variable and a lowercase letter $a$ for its realization. A bold uppercase letter $\boldsymbol{A}$ denotes a random vector and a bold lowercase letter $\boldsymbol{a}$ its realization. For $N \in \mathbb{N}$ realizations of the random vector $\boldsymbol{A}$ we write $\boldsymbol{a}_1, ..., \boldsymbol{a}_N$. The state space of the random vector $\boldsymbol{A}$ is denoted by $\boldsymbol{\mathcal{A}} = \{\boldsymbol{A}(\omega) \,|\, \omega \in \Omega\}$.

Figure 1: Schematic illustration of an optical imaging pipeline, the data states and novel, raw-enabled drift controls. Data $\boldsymbol{x}$ transitions through different representations. The measurement process yields metrologically accurate raw data $\boldsymbol{x}_{\mathrm{RAW}}$, where the errors on each pixel are uncorrelated and unbiased. From the RAW sensor state data undergoes stages of image signal processing (ISP) $\boldsymbol{\Phi}_{\mathrm{Proc}}$, the data model we consider here. Finally, the data is consumed by a machine learning task model $\boldsymbol{\Phi}_{\mathrm{Task}}$ which outputs $\hat{\boldsymbol{y}}$. Combining raw data with the standard machine learning pipeline and a differentiable data model $\boldsymbol{\Phi}_{\mathrm{Proc}}$ enables useful controls for dataset drift comprising ① drift synthesis (creation of physically faithful drift test cases for model selection), ② drift forensics (precise specification of data environments that should be avoided for a given task model), and ③ drift optimization (use of task model gradient to optimize data generating process).

this allows fast and easy generation of test samples. However, [114] point out that perturbations applied to an already processed image can produce drift artifacts that are unfaithful to the physics of camera processing. Results in optics further support the concern that noise obtained from a composite image processing pipeline, such as the data model block in Figure 1, is distinct from noise added to an image that has already been processed on hardware [115, 116]. For illustration, assume we carry out augmentation testing to test the robustness of the task model wrt. to the dataset drift (1). Let $\boldsymbol{\xi} \sim \mathcal{D}_{noise}$ be a noise sample additively applied to the the view resulting in $\boldsymbol{v} + \boldsymbol{\xi}$. Doing so, the task models robustness is tested wrt. the distribution $\mathbb{P} \circ (V + \Xi)^{-1}$ that might not approximate $\mathcal{D}_s$ well. Since $\mathbb{P}$ is unknown, this is difficult to resolve. For physically faithful robustness testing we need to ensure that a sample is an element of the image $\tilde{\boldsymbol{\Phi}}_{\mathrm{Proc}}[\boldsymbol{\mathcal{X}}_{RAW}]$ of $\boldsymbol{\mathcal{X}}_{RAW}$ under $\tilde{\boldsymbol{\Phi}}_{\mathrm{Proc}}$. Accordingly, we define a *physically faithful* data point wrt. the dataset drift (1) as a view $\tilde{\boldsymbol{v}}$ that satisfies $\tilde{\boldsymbol{v}} \in \tilde{\boldsymbol{\Phi}}_{\mathrm{Proc}}[\boldsymbol{\mathcal{X}}_{RAW}]$. In augmentation testing, the test samples are not restricted to physically faithful data points wrt. to any dataset drift, since $\boldsymbol{v} + \boldsymbol{\xi} \in \tilde{\boldsymbol{\Phi}}_{\mathrm{Proc}}[\boldsymbol{\mathcal{X}}_{\mathrm{RAW}}]$ might not hold true for any data model.

A physically faithful alternative to augmentation testing is what we call catalogue testing. It involves the collection of datasets from different cameras which are then used as hold-out robustness validation datasets [49, 117–119]. It does not allow for as flexible and fast in-silico simulation of test cases as augmentation testing because cataloguing requires expensive data collection after which the test cases are "locked-in". Notwithstanding, catalogue testing comes with the appealing guarantee that test samples conform to the processing physics of the different cameras they were gathered from, ensuring that only physically faithful data points are used for testing.

| | Augmentation testing | Catalogue testing | Data models |
|---|:---:|:---:|:---:|
| Simulation of test samples | ✓ | ✗ | ✓ |
| Physically faithful test samples | ✗ | ✓ | ✓ |
| Differentiable data model | ✗ | ✗ | ✓ |

Table 1: A conceptual comparison of different empirical approaches to dataset drift validation for machine learning task models. While augmentation testing allows flexible, ad-hoc synthesis of test cases, they are, in contrast to catalogue testing, not guaranteed to be physically faithful. Pairing qualified raw data with explicit data models allows for flexible synthesis of physically faithful test cases. In addition, the differentiable data model opens up novel drift controls including drift forensics and drift adjustments.

However, the root of input data variations - the data model of images - has received little attention in machine learning robustness research to date. While machine learning practitioners are acutely aware of the dependency between data generation and downstream machine learning model performance, as 75% of respondents to a recent study confirmed [120], data models are routinely treated as a black-box in the robustness literature. This blind spot for explicit data models is particularly surprising since they are standard practice in other scientific communities, in particular optics and metrology [121–124], as well as advanced industry applications, including microscopy [125–127] or autonomous vehicles [128–130].

## 1.2 Our contributions

In this manuscript we outline how to alleviate this blind spot and obtain data models by connecting conventional machine learning with physical optics. Combining raw image data, differentiable data models $\Phi_{\mathrm{Proc}}$ of image signal processing (ISP) and the standard machine learning pipeline enables us to go beyond what is possible with augmentation and catalogue testing. We obtain explicit, differentiable models of the data generating process for flexible, physically faithful drift controls for image machine learning models. Our core contributions are:

- The combination of raw sensor data and data models enables three novel *dataset drift controls*:

  ① **Drift synthesis:** Controlled synthesis of physically faithful drift test cases for model selection and targeted generalization. This is demonstrated for a classification and a regression task and compared to physically unfaithful alternatives (Section 5.1).

  ② **Drift forensics:** Given a particular data model $\Phi_{\mathrm{Proc}}$, the gradient from the upstream task model $\Phi_{\mathrm{Task}}$ can propagate to $\Phi_{\mathrm{Proc}}$, thus enabling precise data forensics: precise specification of data generation environments that should be avoided for a given task model (Section 5.2).

  ③ **Drift optimization:** Lastly, the gradient connection between data $\Phi_{\mathrm{Proc}}$ and task model $\Phi_{\mathrm{Task}}$ allows directly adjusting the data and data generating process themselves via backpropagation to aid the downstream task model learn better faster (Section 5.3).

- We collected and publicly release two raw image datasets in the camera sensor state for data modelling. The raw datasets come with full annotations and processing variations for both a classification (Raw-Microscopy, 17,860 total samples) and a regression (Raw-Drone, 10,412 total samples) task. The data can be downloaded from the anonymized record `https://zenodo.org/record/5235536` as well as through the data loader integrated in our code base below.

- We provide modular PyTorch code for explicit and differentiable data models $\Phi_{\mathrm{Task}}$ from raw camera sensor data. All code is accessible at `https://github.com/aiaudit-org/raw2logit`.

## 1.3 Scope, practical fit and limitation of the proposed methods

The systems infrastructure for optical imaging produced by large industry vendors such as ZEISS, Hamamatsu, Teledyne-Photometrics, Andor, Yokogawa or Perkin-Elmer allow raw sensor readouts. The same applies

to consumer-grade cameras by market-dominating vendors such as Samsung or Apple. Concurrently, ISP-processed data is predominantly used in practice - both in the application domains considered here and machine learning. The reasons are often downstream workload dependencies. In most settings, data is acquired to be human readable for a specific task, for example diagnostics or environmental surveying. Variations in the ISPs, between different vendors or acquisition sites, then lead to the drifts of Equation (1) outlined above. This work is targeted at the current imaging infrastructure that (i) makes widespread use of ISPs that lead to drift and (ii) simultaneously allows access to raw sensor readouts. For this setting, we propose data models that allow engineers to explore, emulate and control different data generating processes related to the ISP at low cost in a physically faithful way. In terms of practical benefits, data models can save time and money (drift synthesis) as additional acquisitions, on the order of days or even weeks, can be avoided. They also open up completely new applications for integrated data-model quality management (drift forensics, drift optimization) which are impossible without differentiable data models. To be clear, there are other important sources of drift, such as the sensor or optical components of the camera, which cannot be captured, yet, by our framework. For example, in the Raw-Microscopy dataset, factors such as the choice of filters for blue, green, and red channels, the point-spread function, and mechanical drift can influence the final image quality. Similarly, in the Raw-Drone dataset, the choice of lens, f-number, PSF circle diameter, ISO, and the gain applied to pixel values can affect the acquired images. These factors introduce variations contributing to dataset drift. The data models presented in this study aim to account for changes during the ISP. While extensions beyond this setting are opportune, raw-only, as sketched out in the drift optimization experiments, would require a shift in the dominant imaging workflows of the application domains considered here. In summary, the current data models offer rich utility to capture ISP as a dominant source of data drift, but are limited to the ISP scope and require an extension to model additional factors outside that scope in a physically faithful manner.

## 2 Related work

While physically sound data models of images have to the best of our knowledge not yet found their way into the machine learning and dataset drift literature, they have been studied in other disciplines, in particular physical optics and metrology. Our ideas on data models and dataset drift controls we present in this manuscript are particularly indebted to the following works.

**Data models for images** [131, 132] employ deep convolutional neural networks for modelling a raw image data processing which is optimized jointly with the task model. In contrast, we employ a parametric data model with tunable parameters that enables the modular drift forensics and synthesis presented later. [133] propose a differentiable image processing pipeline for the purpose of camera lens manufacturing. Their goal, however, is to optimize a physical component (lens) in the image acquisition process and no code or data is publicly available. Existing software packages that provide low level image processing operations include Halide [134], Kornia [135] and the rawpy package [136] which can be integrated with our Python and PyTorch code. We should also take note that outside optical imaging there are areas in machine learning and applied mathematics, in particular inverse problems such as magnetic resonance imaging (MRI) or computed tomography, that make use of known operator learning [137, 138] to incorporate the forward model in the optimization [139] or, as in the case of MRI, learn directly in the k-space [140].

**Drift synthesis** As detailed in Section 1, the synthesis of realistic drift test cases for a task model in computer vision is often done by applying augmentations directly to the input view $v_{GC}$, e.g. a processed `.jpeg` or `.png` image. Hendrycks et al. [43] have done foundational work in this direction developing a practical, standardized benchmark. However, as we explain in Section 1.1 there is no guarantee that noise $\xi$ added to a processed image $v$ will be physically faithful, i.e. that $v + \xi \in \tilde{\Phi}_{\text{Proc}}[\mathcal{X}_{\text{RAW}}]$. This is problematic, as nuances matter [141] for assessing the cascading effects data models have on the task model $\Phi_{\text{Task}}$ downstream [120, 142]. For the same reason, the use of generative models [47] like GANs has been limited for test data generation as they are known to hallucinate visible and less visible artifacts [143, 144]. Other approaches, like the WILDS data catalogue [145, 146], build on manual curation of so called natural distribution shifts, or, like [68], on artificial worst case constructions. These are important tools for the study of dataset drifts, especially those that are created outside the camera image signal processing. Absent explicit, differentiable

data models and raw sensor data, the shared limitation of catalogue approaches is that metrologically faithful drift *synthesis* is not possible and the data generating process cannot be granularly studied and manipulated.

**Drift forensics** Phan et al. [147] use a differentiable raw processing pipeline to propagate the gradient information back to the raw image. Similar to this work, the signal is used for adversarial search. However, Phan et al. optimize adversarial noise on a per-image basis in the raw space $x_{\mathrm{RAW}}$, whereas our work modifies the parameters of the data model $\boldsymbol{\Phi}_{\mathrm{Proc}}$ itself in pursuit of harmful parameter configurations. The goal in this work is not simply to fool a classifier, but to discover failure modes and susceptible parameters in the data model $\boldsymbol{\Phi}_{\mathrm{Proc}}$ that will have the most influence on the task model's performance.

**Drift optimization** An explicit and differentiable image processing data model allows joint optimization together with the task model $\boldsymbol{\Phi}_{\mathrm{Proc}}$. This has been done for radiology image data [148–150] though the measurement process there is different and the focus lies on finding good sampling patterns. For optical data, a related strand of work is modelling inductive biases in the image acquisition process which is explained and contrasted to this work above [116, 133].

**Raw image data** Camera raw files contain the data captured by the camera sensors [121]. In contrast to processed formats such as `.jpeg` or `.png`, raw files contain the sensor data with minimal processing [115, 151, 152]. The processing of the raw data usually differs by camera manufacturer thus contributing to dataset drift. Existing raw data sets from the machine learning, computer vision and optics literature can be organized into two categories. First, datasets that are sometimes treated - usually not by the creators but by users of the data - as raw data but which are in fact not raw. Examples for this category can be found for both modalities considered here [153–163]. All of the preceding examples are processed and stored in formats including `.jpeg`, `.tiff`, `.svs`, `.png`, `.mp4` and `.mov`. Second, datasets that are labelled raw data which are raw. In contrast to the labelled and precisely calibrated raw data presented here, existing raw datasets [164–167] are collected from various sources for image enhancement tasks without full specification of the measurement conditions or labels for classification or segmentation tasks.

## 3 Preliminaries: a data model for images

Before proceeding with a description of the methods we use to obtain the data models $\boldsymbol{\Phi}_{\mathrm{Proc}}$ in this study, let us briefly review the distinction between raw data $x_{\mathrm{RAW}}$, processed image $\boldsymbol{v}$ and the mechanisms $\boldsymbol{\Phi}_{\mathrm{Proc}}{:}\mathbb{R}^{H,W} \to \mathbb{R}^{C,H,W}$ by which image data transitions between these states[3]. Image acquisition has traditionally been optimized for the human perception of a scene [122, 168]. Human eyes detect only the visible spectrum of electromagnetic radiation, hence imaging cameras in different application domains such as medical imaging or remote sensing are usually calibrated to aid the human eye perform a downstream task. This process that gives rise to optical image data, which ultimately forms the backbone for any machine learning downstream, is rarely considered in the machine learning literature. Conversely, most research to date has been conducted on processed RGB image representations. The *raw sensor image* $x_{\mathrm{RAW}}$ obtained from a camera differs substantially from the processed image that is used in conventional machine learning pipelines. The $x_{\mathrm{RAW}}$ state appears like a grey scale image with a grid structure (see $x_{raw}$ in Figure 1). This grid is given by the Bayer color filter mosaic, which lies over sensors [121]. The final *RGB image* $\boldsymbol{v}$ is the result of a series of transformations applied to $x_{\mathrm{RAW}}$. For many steps in this process different possible algorithms exist. Starting from a single $x_{\mathrm{RAW}}$, all those possible combinations can generate an exponential number of possible images that are slightly different in terms of colors, lighting and blur - variations that contribute to dataset drift. In Figure 1 a conventional pipeline from $x_{\mathrm{RAW}}$ to the final RGB image $\boldsymbol{v}$ is depicted. Here, common and core transformations are considered. Note that depending on the application context it is possible to reorder or add additional steps. The symbol $\boldsymbol{\Phi}_i$ is used to denote the $i^{th}$ transformation and $\boldsymbol{v}_i$ (*view*) for the output image of $\boldsymbol{\Phi}_i$. The first step of the pipeline is *black level* correction $\boldsymbol{\Phi}_{\mathrm{BL}}$, which removes any constant offset. The image $\boldsymbol{v}_{\mathrm{BL}}$ is a grey image with a Bayer filter pattern. A *demosaicing* algorithm $\boldsymbol{\Phi}_{\mathrm{DM}}$ is applied to construct the full RGB color image [169]. Given $\boldsymbol{v}_{\mathrm{DM}}$, intensities are adjusted to obtain a neutrally illuminated image $\boldsymbol{v}_{\mathrm{WB}}$ through a *white balance* transformation $\boldsymbol{\Phi}_{\mathrm{WB}}$. By considering color dependencies, a *color correction* transformation $\boldsymbol{\Phi}_{\mathrm{CC}}$ is applied to balance hue and saturation of the image. Once lighting and colors are corrected, a *sharpening* algorithm $\boldsymbol{\Phi}_{\mathrm{SH}}$ is

---

[3]We recommend [122] for a good introduction to the physics of digital optical imaging.

applied to reduce image blurriness. This transformation can make the image appear more noisy. For this reason a *denoising* algorithm $\Phi_{DN}$ is applied afterwards [170, 171]. Finally, *gamma correction*, $\Phi_{GC}$, adjusts the linearity of the pixel values. For a closed form description of these transformations see Section 4.2. Compression may also take place as an additional step. It is not considered here as the input image size is already small. Furthermore, the effect of compression on downstream task model performance has been thoroughly examined before [172–176]. However, users of our code can add this step or reorder the sequence of steps in the modular processing object class per their use case needs[4].

## 4    Methods

In order to obtain advanced data models for images, raw sensor data is required. In many industry domains such as microscopy, biomedicine, autonomous vehicles or remote sensing raw sensor data is processed at scale for machine vision tasks. Most existing digital camera systems on the market today, including consumer smartphones, can be configured to access the raw sensor measurements. Next, we explain how to obtain raw sensor data from existing optical hardware. We collected two datasets for two representative machine learning tasks. Both datasets are made available for free, public use at `https://zenodo.org/record/5235536`.

### 4.1    Raw dataset acquisition

As public, scientifically calibrated and labelled raw data is, to the best of our knowledge, currently not available, we acquired two raw datasets as part of this study: Raw-Microscopy and Raw-Drone. Raw-Microscopy consists of expert annotated blood smear microscope images. Raw-Drone comprises drone images with annotations of cars. Our motivation behind the acquisition of these particular datasets was threefold. First, we wanted to ensure that the acquired datasets provide good coverage of representative machine learning tasks, including classification (Raw-Microscopy) and regression (Raw-Drone). Second, we wanted to collect data on applications that, to our minds, are disposed towards positive welfare impact in today's world, including medicine (Raw-Microscopy) and environmental surveying (Raw-Drone). Third, we wanted to ensure the downstream machine learning application contexts are such where errors can be costly, here patient safety (Raw-Microscopy) and autonomous vehicles (Raw-Drone), hence necessitating extensive robustness and dataset drift controls. Since data collection is an expensive project in and of itself we did not aspire to provide extensive benchmark datasets for the respective applications, but to collect enough data to demonstrate the advanced data modelling and dataset drift controls that raw data enables.

In Appendix A.4.1 we provide detailed information on the two datasets and the calibration setups of the acquisition process. Samples of both datasets can be inspected in Figure 2 and Appendix A.4.1. Full datasheet documentation following [177] is also available in Appendix A.4.2.

An alternative approach could be to attempt reconstruct raw images from processed images [178, 179]. As we laid out earlier, when an image is captured by a digital camera, the sensor records raw image data in the form of an array of pixel values. This raw image data is usually processed by an ISP before being used in a downstream task. The ISP performs various adjustments such as color correction, noise reduction, and sharpening to enhance the quality of the image. However, these adjustments are not physically faithful to the original raw image data and result in a loss of information. Therefore, it is generally not possible to reconstruct the exact raw image data from the ISP processed images. While the processed images may look better to the human eye, they do not accurately represent the physical reality of the original scene. For example, a recent paper by Nam et al. [179] propose a content-aware metadata approach to sRGB-to-Raw RGB de-rendering, but acknowledges that the resulting approximations are not perfectly accurate and still suffer from limitations. Similarly, another study by Brooks et al. [178] present a method for recovering raw data from processed images, but also note that the approach is not able to recover all of the original data with perfect accuracy. These findings highlight the fundamental challenge of reconstructing raw data from processed images. Empirical approximations are possible but not exact, that is not physically faithful, and hence orthogonal to our goal here. However, one should note that these reconstruction approaches can offer

---

[4]See `pipeline_torch.py` and `pipeline_numpy.py` in our code.

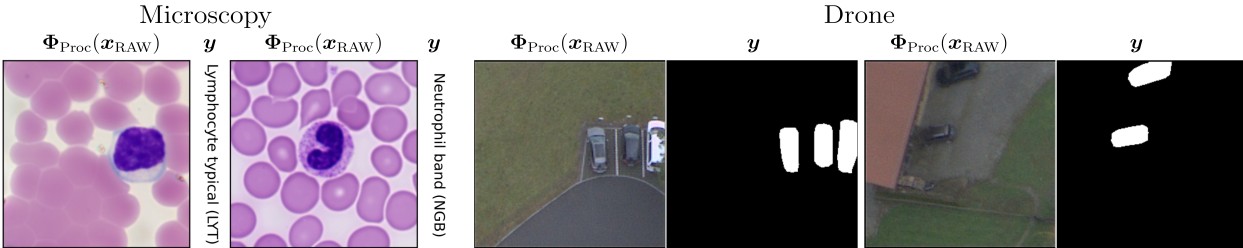

Figure 2: Processed samples and labels of the two datasets, Raw-Microscopy (columns one to four) and Raw-Drone (columns five and eight), that were acquired for the dataset drift study presented here.

interesting value propositions outside the physically precise drift regime. The proposed technique by [178] "unprocesses" images and offers interesting gains during training, enabling a convolutional neural network with 14-38% lower error rates and 9-18x faster performance, while generalizing to other sensors as well. This approach can further be calibrated using joint learning of sampling and reconstruction, offering better raw reconstructions by adapting to image content, with an additional online fine-tuning strategy for enhanced results [179].

## 4.2  Data models: Image signal processing $\Phi_{\text{Proc}}$

The second ingredient to this study are the data models of image processing. Let $(\boldsymbol{X}_{RAW}, Y) : \Omega \to \mathbb{R}^{H,W} \times \mathcal{Y}$ be the raw sensor data generating random variable on some probability space $(\Omega, \mathcal{F}, \mathbb{P})$, with $\mathcal{Y} = \{0,1\}^K$ for classification and $\mathcal{Y} = \{0,1\}^{H,W}$ for segmentation. Let $\boldsymbol{\Phi}_{\text{Task}} : \mathbb{R}^{C,H,W} \to \mathcal{Y}$ be the task model determined during training. The inputs that are given to the task model $\boldsymbol{\Phi}_{\text{Task}}$ are the outputs of the data model $\boldsymbol{\Phi}_{\text{Proc}}$. We distinguish between the raw sensor image $\boldsymbol{x}_{\text{RAW}}$ and a *view* $\boldsymbol{v} = \boldsymbol{\Phi}_{\text{Proc}}(\boldsymbol{x}_{\text{RAW}})$ of this image, where $\boldsymbol{\Phi}_{\text{Proc}} : \mathbb{R}^{H,W} \to \mathbb{R}^{C,H,W}$ models the transformation steps applied to the raw sensor image during processing.

The objective in supervised machine learning is to learn a task model $\boldsymbol{\Phi}_{\text{Task}} : \mathbb{R}^{C,H,W} \to \mathcal{Y}$ within a fixed class of task models $\mathcal{H}$ that minimizes the expected loss wrt. the loss function $\mathcal{L} : \mathcal{Y} \times \mathcal{Y} \to [0, \infty)$, that is to find $\boldsymbol{\Phi}_{\text{Task}}^{\star}$ such that

$$\inf_{\boldsymbol{\Phi}_{\text{Task}} \in \mathcal{H}} \mathbb{E}[\mathcal{L}(\boldsymbol{\Phi}_{\text{Task}}(\boldsymbol{V}), Y)] \tag{2}$$

is attained. Towards that goal, $\boldsymbol{\Phi}_{\text{Task}}$ is determined during training such that the empirical error

$$\frac{1}{N} \sum_{n=1}^{N} \mathcal{L}(\boldsymbol{\Phi}_{\text{Task}}(\boldsymbol{v}_n), y_n) \tag{3}$$

is minimized over a sample $\mathcal{S} = ((\boldsymbol{v}_1, y_1), ..., (\boldsymbol{v}_N, y_N))$ of views. Modelling in the conventional machine learning setting begins with the image data generating random variable $(\boldsymbol{V}, Y) = (\boldsymbol{\Phi}_{\text{Proc}}(\boldsymbol{X}_{RAW}), Y)$ and the target distribution $\mathcal{D}_t = \mathbb{P} \circ (\boldsymbol{V}, Y)^{-1}$. Given a dataset drift $\mathcal{D}_s = \mathbb{P} \circ (\tilde{\boldsymbol{V}}, Y)^{-1} \neq \mathcal{D}_t$, as specified in Equation (1), without a data model we have little recourse to disentangle reasons for performance drops in $\boldsymbol{\Phi}_{\text{Task}}$. To alleviate this underspecification, an explicit data model is needed. We consider two such models in this study: a static model $\boldsymbol{\Phi}_{\text{Proc}}^{\text{stat}}$ and a parametrized model $\boldsymbol{\Phi}_{\text{Proc}}^{\text{para}}$.

In the following, we denote by $\boldsymbol{x}_{\text{RAW}} \in [0,1]^{H,W}$ the normalized raw image, that is a grey scale image with a Bayer filter pattern normalized by $2^{16} - 1$, i.e.

$$\boldsymbol{x}_{\text{RAW}} = \begin{bmatrix} \boldsymbol{A}_{1,1} & . & . & . & \boldsymbol{A}_{1,\frac{W}{2}} \\ . & . & & & . \\ . & & . & & . \\ . & & & . & . \\ \boldsymbol{A}_{\frac{H}{2},1} & . & . & . & \boldsymbol{A}_{\frac{H}{2},\frac{W}{2}} \end{bmatrix} \quad with \quad \boldsymbol{A}_{h,j} = \begin{bmatrix} r_{2h+1,2w+1} & g_{2h+1,2w} \\ g_{2h,2w+1} & b_{2h,2w} \end{bmatrix}, \tag{4}$$

where the values $r_{2h+1,2w+1}, g_{2h+1,2w}, g_{2h,2w+1}, b_{2h,2w}$ correspond to the values measured through the different sensors and normalized by $2^{16} - 1$. We provide here a precise description of the transformations that we consider

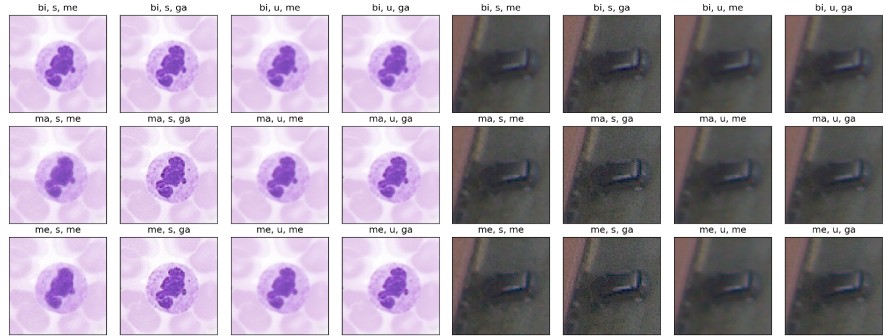

| Data models | Used functions | | |
|---|---|---|---|
| bi,s,me | $\mathbf{\Phi}_{\text{DM}}^{\text{Bil}}$ | $\mathbf{\Phi}_{\text{SH}}^{\text{SF}}$ | $\mathbf{\Phi}_{\text{DN}}^{\text{MD}}$ |
| bi,s,ga | $\mathbf{\Phi}_{\text{DM}}^{\text{Bil}}$ | $\mathbf{\Phi}_{\text{SH}}^{\text{SF}}$ | $\mathbf{\Phi}_{\text{DN}}^{\text{GD}}$ |
| bi,u,me | $\mathbf{\Phi}_{\text{DM}}^{\text{Bil}}$ | $\mathbf{\Phi}_{\text{SH}}^{\text{UM}}$ | $\mathbf{\Phi}_{\text{DN}}^{\text{MD}}$ |
| bi,u,ga | $\mathbf{\Phi}_{\text{DM}}^{\text{Bil}}$ | $\mathbf{\Phi}_{\text{SH}}^{\text{UM}}$ | $\mathbf{\Phi}_{\text{DN}}^{\text{GD}}$ |
| me,s,me | $\mathbf{\Phi}_{\text{DM}}^{\text{Men}}$ | $\mathbf{\Phi}_{\text{SH}}^{\text{SF}}$ | $\mathbf{\Phi}_{\text{DN}}^{\text{MD}}$ |
| me,s,ga | $\mathbf{\Phi}_{\text{DM}}^{\text{Men}}$ | $\mathbf{\Phi}_{\text{SH}}^{\text{SF}}$ | $\mathbf{\Phi}_{\text{DN}}^{\text{GD}}$ |
| me,u,me | $\mathbf{\Phi}_{\text{DM}}^{\text{Men}}$ | $\mathbf{\Phi}_{\text{SH}}^{\text{UM}}$ | $\mathbf{\Phi}_{\text{DN}}^{\text{MD}}$ |
| me,u,ga | $\mathbf{\Phi}_{\text{DM}}^{\text{Men}}$ | $\mathbf{\Phi}_{\text{SH}}^{\text{UM}}$ | $\mathbf{\Phi}_{\text{DN}}^{\text{GD}}$ |
| ma,s,me | $\mathbf{\Phi}_{\text{DM}}^{\text{Mal}}$ | $\mathbf{\Phi}_{\text{SH}}^{\text{SF}}$ | $\mathbf{\Phi}_{\text{DN}}^{\text{MD}}$ |
| ma,s,ga | $\mathbf{\Phi}_{\text{DM}}^{\text{Mal}}$ | $\mathbf{\Phi}_{\text{SH}}^{\text{SF}}$ | $\mathbf{\Phi}_{\text{DN}}^{\text{GD}}$ |
| ma,u,me | $\mathbf{\Phi}_{\text{DM}}^{\text{Mal}}$ | $\mathbf{\Phi}_{\text{SH}}^{\text{UM}}$ | $\mathbf{\Phi}_{\text{DN}}^{\text{MD}}$ |
| ma,u,ga | $\mathbf{\Phi}_{\text{DM}}^{\text{Mal}}$ | $\mathbf{\Phi}_{\text{SH}}^{\text{UM}}$ | $\mathbf{\Phi}_{\text{DN}}^{\text{GD}}$ |

(a) Samples for both datasets, Raw-Microscopy and Raw-Drone, from all twelve static data models $\mathbf{\Phi}_{\text{Proc}}^{\text{stat}}$ used for the drift synthesis experiments in Section 5.1. A version with higher resolution is omitted here to save space and can instead be found in Figure 8 in the appendices.

(b) Abbreviations of the twelve configurations of the static data model $\mathbf{\Phi}_{\text{Proc}}^{\text{stat}}$ used in the drift synthesis experiments.

Figure 3: Samples for both datasets (a) and abbreviations of the twelve data models (b)

in our static model $\mathbf{\Phi}_{\text{Proc}}^{\text{stat}}$, followed by a description how to convert this static model into a differentiable, parametrized model $\mathbf{\Phi}_{\text{Proc}}^{\text{para}}$.

### 4.2.1 The static data model $\Phi_{\text{Proc}}^{\text{stat}}$

Following common steps in ISP, the *static data model* is defined as the composition

$$\mathbf{\Phi}_{\text{Proc}}^{\text{stat}} = \mathbf{\Phi}_{\text{GC}} \circ \mathbf{\Phi}_{\text{DN}} \circ \mathbf{\Phi}_{\text{SH}} \circ \mathbf{\Phi}_{\text{CC}} \circ \mathbf{\Phi}_{\text{WB}} \circ \mathbf{\Phi}_{\text{DM}} \circ \mathbf{\Phi}_{\text{BL}}, \tag{5}$$

mapping a raw sensor image to a RGB image. We note that other data model variations, for example by reordering or adding steps, are feasible. The static data models allow the controlled synthesis of different, physically faithful views from the same underlying raw sensor data by manually changing the configurations of the intermediate steps. Fixing the continuous features, but varying $\mathbf{\Phi}_{\text{DM}}$, $\mathbf{\Phi}_{\text{SH}}$ and $\mathbf{\Phi}_{\text{DN}}$ results in twelve different views for the configurations considered here. Samples for each of the twelve data models are provided in Figure 3a. The individual functions of the composition $\mathbf{\Phi}_{\text{Proc}}^{\text{stat}}$ can be found in Appendix A.1.1. An overview of the data model configurations and their corresponding abbreviations can be found alongside processed samples in Figures 3a and 3b.

### 4.2.2 The parametrized data model $\Phi_{\text{Proc}}^{\text{para}}$

For a fixed raw sensor image, the *parametrized data model* $\mathbf{\Phi}_{\text{Proc}}^{\text{para}}$ maps from a parameter space $\Theta$ to a RGB image. It is similar to the static data model with the notable difference that each processing step is differentiable wrt. its parameters $\boldsymbol{\theta}$. This allows for backpropagation of the gradient from the output of the task model $\mathbf{\Phi}_{\text{Task}}$ through the data model $\mathbf{\Phi}_{\text{Proc}}$ all the way back to the raw sensor image $\boldsymbol{x}_{\text{RAW}}$ to perform drift forensics and drift adjustments. Hence, we aim to design a data model $\mathbf{\Phi}_{\text{Proc}}^{\text{para}} : \mathbb{R}^{H,W} \times \Theta \to \mathbb{R}^{C,H,W}$ that is differentiable in $\boldsymbol{\theta} \in \Theta$ satisfying

$$\mathbf{\Phi}_{\text{Proc}}^{\text{stat}} = \mathbf{\Phi}_{\text{Proc}}^{\text{para}}\left(\cdot, \boldsymbol{\theta}^{\text{stat}}\right) \tag{6}$$

for some choice of parameters $\boldsymbol{\theta}^{\text{stat}}$ and some fixed configuration of the static pipeline $\mathbf{\Phi}_{\text{Proc}}^{\text{stat}}$. Using the individual functional components specified in Appendix A.1.2, we define for $\boldsymbol{\theta} = (\boldsymbol{\theta}_1, ..., \boldsymbol{\theta}_7) \in \Theta$ the parametrized processing model

$$\mathbf{\Phi}_{Proc}^{\text{para}} : [0,1]^{3,H,W} \times \Theta \to [0,1]^{3,H,W}, (\boldsymbol{x}_{\text{RAW}}, \boldsymbol{\theta}) \mapsto \boldsymbol{v} \tag{7}$$

by the composition

$$\boldsymbol{v} = \left(\mathbf{\Phi}_{\text{GC}}^{\text{para}}(\cdot, \boldsymbol{\theta}_7) \circ \mathbf{\Phi}_{\text{DN}}^{\text{para}}(\cdot, \boldsymbol{\theta}_6) \circ \mathbf{\Phi}_{\text{SH}}^{\text{para}}(\cdot, \boldsymbol{\theta}_5) \circ \mathbf{\Phi}_{\text{CC}}^{\text{para}}(\cdot, \boldsymbol{\theta}_4) \circ \mathbf{\Phi}_{\text{WB}}^{\text{para}}(\cdot, \boldsymbol{\theta}_3) \circ \mathbf{\Phi}_{\text{DM}}^{\text{para}}(\cdot, \boldsymbol{\theta}_2) \circ \mathbf{\Phi}_{\text{BL}}^{\text{para}}(\cdot, \boldsymbol{\theta}_1)\right)(\boldsymbol{x}_{\text{RAW}}). \tag{8}$$

The operations used above are differentiable except for the clipping operation in the GC that is *a.e.*-differentiable[5], since the set $\{0,1\}$ of non-differentiable points has measure zero. Assuming in addition that $\mathbb{P}\left((v_{\text{DN}})_{c,h,w} \in \{0,1\}\right) = 0$ holds true for the entries of $\boldsymbol{v}_{\text{DN}}$ results in an *a.e.*-differentiable processing

---

[5] *a.e.* stands for almost everywhere

model. We further say that $\mathbf{\Phi}_{\mathrm{Proc}}^{\mathrm{para}}$ is differentiable, noting that this holds only *a.e.* under the aforementioned assumption.

### 4.3  Task models $\Phi_{\mathbf{Task}}$

Finally, two task models are employed in the experiments. For the classification task on the Raw-Microscopy dataset a 18-layer residual net (ResNet18) [180] was used as reference task model. To segment cars from the Raw-Drone dataset the convolutional neural network proposed in [181] (U-Net) was used. Both task models were trained using common data augmentations on processed views $\boldsymbol{v}$ of the image measurements to avoid naive robustness failures. A detailed description of the task models and their hyperparameters is given in A.2.

## 5  Applications

With data models, raw data and task models in place we are now able to demonstrate the advanced dataset drift controls comprising ① drift synthesis, ② modular drift forensics and ③ drift optimization.

### 5.1  Drift synthesis

The static data model enables physically faithful synthesis of drift test cases: individual components of the data model can be swapped out, allowing the controlled creation of different, physically faithful processed views from one raw reference dataset. A typical use case of drift synthesis for machine learning researchers and practitioners is the prospective validation of their task model to drift from different camera devices, for example microscopes across different labs, without having to collect measurements from the different devices. This simulation can be done insilico as software because the hardware specific processing that takes place on optical measurement devices after the sensor reading is also insilico. Thus, the extraction of raw sensor readings from one device allows the emulation of different processing variations present on other devices. A typical workflow of this data synthesis starts with an engineer constructing a data model of interest, then passing raw measurements through it and finally getting emulated data to test how the downstream task model would fare on processing variations from different devices.

We provide twelve possible example data models in the following experiments. For each of the twelve data models laid out in Section 4.2, the task models were trained for 100 epochs on image data processed through the training data model. Hyperparameters were kept constant across all runs to isolate the effect of varying the data models. Then, dataset drift test cases were synthesized by processing the raw test data through the remaining eleven data models. The task models were then evaluated on test data from all twelve data models. All results that follow are reported as the mean with error bars over a 5-fold cross-validation[6]. The metrics used to evaluate the task models are accuracy for classification and IoU for segmentation.

#### 5.1.1  Physically faithful versus physically unfaithful robustness validation

The leukocyte classification model, as displayed in the left matrix of Figure 4, has a critical drop for few configurations, suggesting that it is relatively robust to processing induced dataset drift except for the (ma,s,me) configuration. Note that diagonal elements serve as reference corresponding to test data that was processed in the same way as the training data. The segmentation task model (left matrix in Figure 5) displays a more heterogeneous pattern with symmetries for certain combinations of data models, such as (bi, u, me/ga) and (me, s, me/ga), which are mutually destructive to the task model performance. The average performance of the task models drops from 0.82 to 0.8 between train and test data models for classification and from 0.71 to 0.65 for segmentation. That is the average change from train to test data environment calculated across all configurations for ISP as well as Common Corruptions. The results for individual components of the data models can also be directly compared in Figures 4 and 5. For example, to understand how changes in the demosaicing algorithm affect the segmentation model, we can look at the left box in Figure 5 and focus on the column combinations 1-5-9, 2-6-10, 3-7-11 and 4-8-12 where the demosaicing is varied but the other components of the data model stay fixed. Considering the training condition with the

---

[6]You can find a full description of task model hyperparameters and experimental setup in Appendix A.2.

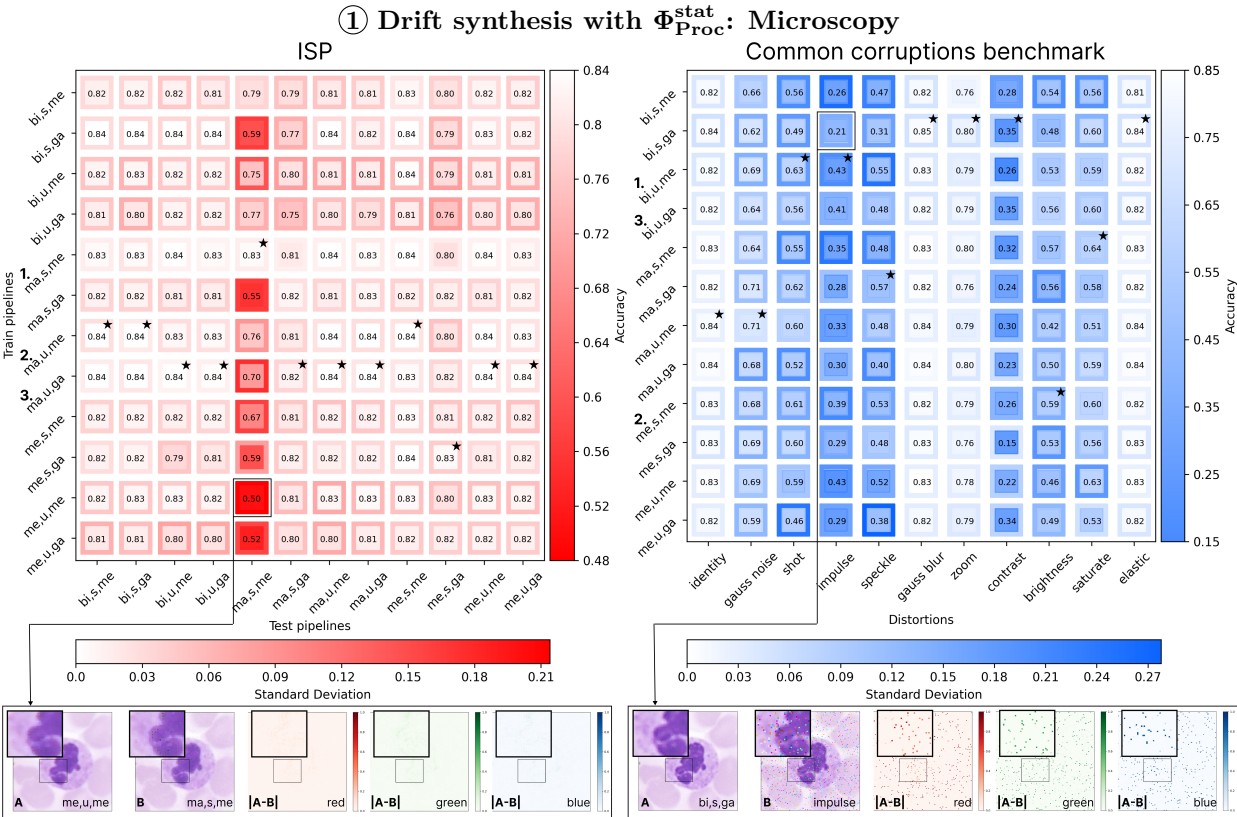

Figure 4: 5-fold cross-validation results of the Raw-Microscopy drift synthesis experiments. Each cell contains the average accuracy with a color coded border for the standard deviation. Task models were trained on the data models on the vertical axis and then tested on processed data as indicated on the horizontal axis. Numbers 1-3 left to the vertical axis denote the ranking of task models according to their average accuracy across all test pipelines (respective corruptions). Stars denote the train pipeline under which the task model performed best on the respective test pipeline/corruption. Full ranking results can be found in Tables 5 to 7 of Appendix A.3. Top-left: Varying the data model leads to mild performance drops except (ma,s,me). Diagonal is $\Phi_{\mathrm{Proc}} = \tilde{\Phi}_{\mathrm{Proc}}$. Top-Right: Comparison to the corruption benchmark at medium severity (level 3). The average performance drop is more than thirteen times higher compared to data model variations. First column is $\Phi_{\mathrm{Proc}} = \tilde{\Phi}_{\mathrm{Proc}}$. Bottom: Visual inspection of worst case (globally worst scoring) train/test pipelines.

(bi,s,me) data model using Bilinear demosaicing (row 1), the task model performance drops from 0.7 (column 1) to 0.67 (column 5) IoU in response to Malvar2004 demosaicing and to 0.66 (column 9) IoU when using Menon2007.

To demonstrate the limitation of post-hoc augmentations[7], we compare the drift synthesis results to a popular augmentation testing framework known as Common Corruptions Benchmark [43]. In machine learning practice, augmentation users often assume that applying a corruption, for example 'blur', to a processed image will emulate the noise from a real-world camera system, for example blur from the lens or the denoising component in the camera. However, this is not the case. It should not come as a surprise given the composition of the optical data generating process (see Figure 1 and Section 4.2), that is $\boldsymbol{v} + \boldsymbol{\xi} \in \tilde{\Phi}_{\mathrm{Proc}}[\boldsymbol{\mathcal{X}}_{\mathrm{RAW}}]$ might not hold true for any data model as we explain in Section 1.1. This has also been empirically demonstrated in previous work [114–116]. Here we go one step further to show that physically *un*faithful augmentation testing can lead to wrong conclusions in model selection.

---

[7]We are only referring to limitations relating to robustness testing and model selection. Augmentations have important empirically validated benefits in other applications such as regularization during training or semi- and self-supervised learning.

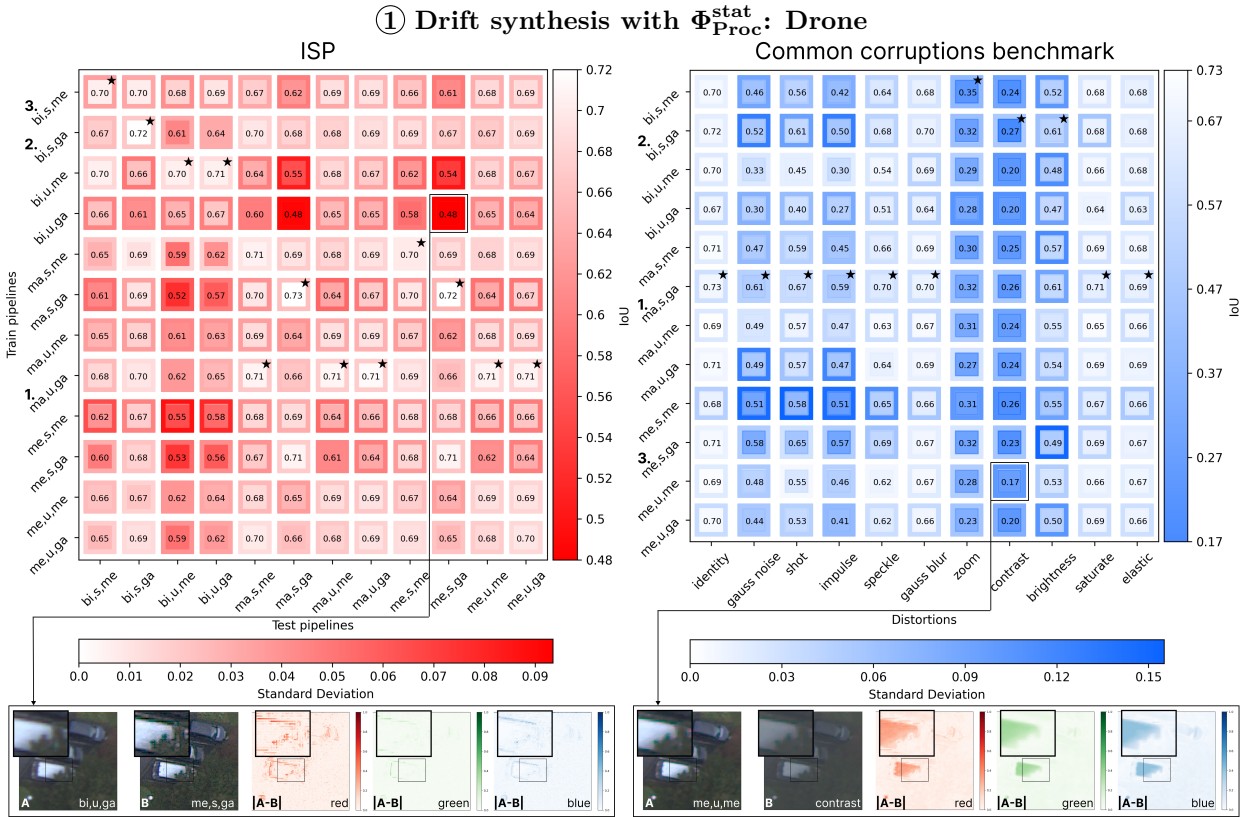

Figure 5: 5-fold cross-validation results of the Raw-Drone drift synthesis experiments. Each cell contains the average IoU with a color coded border for the standard deviation. Task models were trained on the data model on the vertical axis and then tested on processed data as indicated on the horizontal axis. Numbers 1-3 left to the vertical axis denote the ranking of task models according to their average IoU across all test pipelines respective corruptions. Stars denote the train pipeline under which the task model performed best on the respective test pipeline/corruption. Full ranking results can be found in Tables 5, 8 and 9 of Appendix A.3. Left: Varying the data model leads to mixed performance drops. Diagonal is $\mathbf{\Phi}_{\mathrm{Proc}} = \tilde{\mathbf{\Phi}}_{\mathrm{Proc}}$. Right: Comparison to the corruption benchmark at medium severity (level 3). The average performance drop is more than four times higher compared to data model variations. First column is $\mathbf{\Phi}_{\mathrm{Proc}} = \tilde{\mathbf{\Phi}}_{\mathrm{Proc}}$. Bottom: Visual inspection of worst case (globally worst scoring) train/test pipelines.

As we note in Appendix A.3, a direct apple-to-apple comparison is impossible due to the fundamental limitation of post-hoc corruptions' physical unfaithfulness. However, we make the comparison as plausible as possible by only including corruptions that can be related to the ISP data model. Others, such as Fog, Spatter, Motion, Snow, Frost were excluded[8]. In contrast to physically faithful test data, the performance drops under corruptions are more severe across the board: from 0.82 to 0.55 for classification and from 0.71 to 0.49 for segmentation[9]. This is more than thirteen and four times as much as for the physically faithful drifts synthesized with the data models considered here. We see similar gaps when considering the best models. For example, the best performing microscopy data-task-model combo selected across all test ISPs ((ma,s,me) with 0.83 average accuracy) has more than 20 percentage points gap compared to Common Corruptions (0.62 average accuracy). For the segmentation task we make a similar observation where the best performing drone data-task-model combo selected across all test ISPs ((ma,u,ga) with 0.68 average IoU) has more than 10 percentage points gap compared to Common Corruptions (0.55 average IoU) (see Appendix A.3 for full tables.) The qualitative difference between physically faithful drift test cases and augmentation testing can also be appreciated in the samples of the bottom rows of Figures 4 and 5. For each task we display a sample

---

[8]A comparative overview of included and excluded corruptions can be found in Figure 13 of Appendix A.3

[9]Results at additional severity levels for the common corruptions can be found in Appendix A.3.

from the drift test configuration with the worst case performance drop between train and test data conditions. We show the sample viewed from training data model ($\mathbf{A}$), the test data model ($\mathbf{B}$), and the difference between both ($|\mathbf{A}\text{-}\mathbf{B}|$) along the red, green and blue channel. For both tasks, the drift artifacts ($|\mathbf{A}\text{-}\mathbf{B}|$) are more localized than the artifacts obtained from augmentation testing. This makes sense, as changes in the composition of the test data models $\mathbf{\Phi}_{\text{Proc}}$ maintain the physical faithfulness of the remaining data model, whereas augmentation testing spreads noise globally across all pixels which is not guaranteed to be physically faithful.

### 5.1.2 Implications for model selection

Similarly, the conclusions for model selection diverge depending on whether physically faithful data or corruptions are used. In terms of the average performance across all test conditions, none of the top-3 training data models[10] overlap between ISP and common corruptions on the classification task. For segmentation, only one of the training data models (bi,s,ga) overlaps in the top-3 under ISP and common corruptions. Similarly, the training data models under which task models perform best in individual testing conditions vary widely between ISP and common corruptions, both for classification and segmentation. Why does physical faithfulness matter in dataset drift testing? A test result is only as reliable as its constituting parts. If we are to rely on robustness test results to decide whether to use a task model in a certain data environment or not, we need to ensure the test cases represent real-world data models. If the test cases are not physically faithful, the results based on them are of limited use to make decisions.

### 5.1.3 Data models and targeted generalization

Recent advances in learning theory by Krikamol Muandet conjecture the impossibility to design rational learning algorithms that have the ability to successfully learn across heterogeneous environments [96]. Explicit data models allow us to rethink the problem of generalization in a similar vein. With data models it is possible to i) precisely specify individual environments and ii) observe what combinations of environments and task model play together nicely. Rather than selecting task models with best average performance across all heterogeneous test environments, we can serve the task model with the right data model depending on which environment it is deployed in. When we analyze the columns of the matrices in Figures 4 and 5 we can observe under what training data model (or 'environments' in Muandet's language) the task model performs best in which testing environment. These configurations are marked by a star ($\star$). For example, in the case of classification we can observe that for a task model to perform well in (bi,u,me) and (bi,u,ga) test data environments, the (ma,u,ga) training data environment is best (left matrix, Figure 4). However, for the segmentation task, to perform best in the same testing data environments, the (bi,u,me) training data environment is preferable (left matrix, Figure 5).

### 5.1.4 Use cases and limitations of drift synthesis

The most immediate use cases for drift synthesis is physically faithful, prospective validation without measurement. In this scenario an engineer will have a task model as well as reference raw measurements. She would then construct data models of interest, for example for two different microscopes across laboratory sites $s$ and $t$. She would then pipe the reference measurements through data models $s$ and $t$ to obtain two different datasets and test the task models on them. She could observe the effect of the processing in lab site $t$ from a computer without ever having to take expensive measurements on site. Building up a catalogue of data models, as we demonstrated in the experiments, then further allows to perform model selection or targeted generalization management where the task model is paired with suitable data models during deployment. All these applications presuppose access to raw data as well as knowledge of the data model specification so that they can be constructed accordingly in software.

## 5.2 Drift forensics

Clear and precise specification of the limitations of use is a mandated requirement for many products that can potentially contain machine learning components, such as software as a medical device [105, 106] or

---

[10]Denoted by the numbers 1-3 alongside the rows of the matrices in Figures 4 and 5

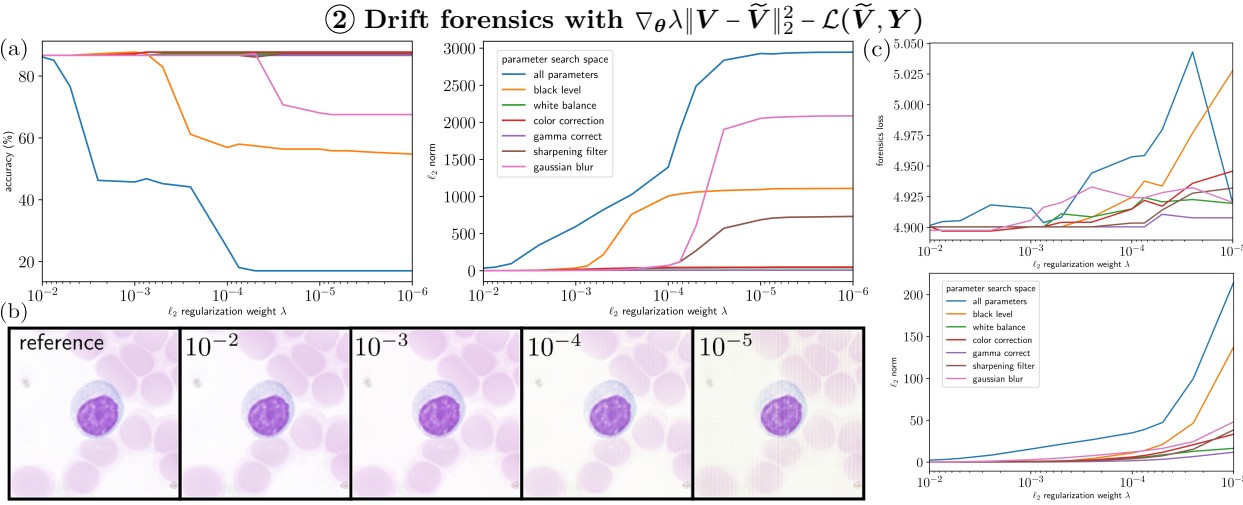

Figure 6: (a): Test accuracy on the Raw-Microscopy test set after 20 epochs of adversarial search in the data model for varying regularization weight parameters $\lambda$. The individual plots depict the various pipeline parameter selections (left plot). Plot showing $\ell_2$-norm (of processed images between the adversarially trained $\widetilde{\mathbf{\Phi}}_{\mathrm{Proc}}^{\mathrm{para}}$ and the default $\mathbf{\Phi}_{\mathrm{Proc}}^{\mathrm{para}}$) versus attained accuracy of the task model (right plot). The metrics are evaluated on the test set after 20 epochs of adversarial optimization for varying regularization weight parameter $\lambda$. (c): Same for Raw-Drone. The individual plots depict the various data model parameter selections. A lower regularization results in a bigger search space for adversarial optimization. Forensics loss refers to the binary cross entropy and Dice loss used as the optimization objective for the segmentation task model. (b): Processed samples from the drift forensics after 20 epochs with varying regularization weights $\lambda$.

autonomous vehicles [182]. Without knowledge and control over the data acquisition process in practice this requirement cannot be met. An explicit, differentiable data model paired with raw data offers a viable solution to this problem. $\mathbf{\Phi}_{\mathrm{Proc}}^{\mathrm{para}}$ enables the analysis of the task model's susceptibility to dataset drift in an interpretable manner using adversarial search. Related work, such as [147] also uses a differentiable raw processing pipeline to propagate the gradient information back to the raw image. There, however, the signal is used in a classical adversarial setup, to optimize adversarial noise on a per-image basis. In our work here, gradient updates are not applied to individual images, but to the data model parameters. The goal of such an analysis is to identify the parameter configurations of the data model under which the task model should not be operated. The resulting adjustments correspond to plausible changes which reflect changes in data model, for example due to changing camera ISPs. In order to limit the parameter ranges, we chose an explicit constraint in the RGB space.

$$\underset{\widetilde{\boldsymbol{\theta}} \in \Theta}{\text{minimize}} \quad \lambda \|\boldsymbol{V} - \widetilde{\boldsymbol{V}}\|_2^2 - \mathcal{L}(\widetilde{\boldsymbol{V}}, \boldsymbol{Y}), \tag{9}$$

where $\boldsymbol{V} = \mathbf{\Phi}_{\mathrm{Proc}}^{\mathrm{para}}(\boldsymbol{X}_{\mathrm{RAW}}, \boldsymbol{\theta})$ are the RGB images obtained from the original data model and $\widetilde{\boldsymbol{V}} = \mathbf{\Phi}_{\mathrm{Proc}}^{\mathrm{para}}(\boldsymbol{X}_{\mathrm{RAW}}, \widetilde{\boldsymbol{\theta}})$ are the RGB images obtained from adversarial search on the data model parameters. Equation (9) maximizes the classification loss under a relaxed $\ell_2$-constraint controlled by the hyperparameter $\lambda \geq 0$. This procedure yields data model parameters that deteriorate the task model performance while keeping the measured distortion minimal and the within constraints of physical faithfulness. All of the pipeline's parameters are optimized jointly to search for a task model's overall data model related weaknesses. Targeting select parameters is also possible and provides insight into a parameter's effect on the task model's performance.

### 5.2.1 Sensitivity to data models

The left plot in block (a) of Figure 6 shows sensitivities of the classification task model to changes in the data model parameters. With increased relaxation of the $\ell_2$-regularization, the accuracy declines exposing configurations under which the task model deteriorates. As to be expected, the setting allowing for all

parameters to be altered shows the biggest effect on the resulting performance. Individually, changes in the black level configuration $\Phi_{\text{BL}}^{para}$ and the denoising parameters $\Phi_{\text{DN}}^{para}$ pose the greatest risk for task model performance under a relaxed regularization weight. In contrast to the classification model the performance drops for the segmentation model are less severe (top plot in block (c) of Figure 6). We hypothesize that this is because classification problems are inherently discontinuous while inverse problems inherently allow for more stable solutions [183], thus being less susceptible to instabilities.

### 5.2.2 Sensitivity in relation to magnitude

For comparison, the right plot in block (a) of Figure 6 shows the regularization weight $\lambda$ against the resulting $\ell_2$. Interestingly, a higher norm in the resulting RGB images does not directly translate to the most severe performance degradation of the task model. At $\ell_2 = 10^{-5}$, changes in the Gaussian blur parameters induce a norm almost twice as large as the changes in the black level parameters. However, the corresponding drop in accuracy caused by Gaussian blur is around one third less relative to black level. Similarly, at $\ell_2 = 10^{-5}$, the sharpening filter parameters incur a norm but do not lead to accuracy drops of the task model. This underscores the importance of precise data models for dataset drift validation. Physically faithful yet small changes, as visible in the samples in bottom row of Figure 6, in processed images can have larger impact on the performance than large changes.

### 5.2.3 Use cases and limitations of drift forensics

A practical use-case of drift forensics looks follows: party $\boldsymbol{s}$ develops and trains a model and then licenses it to party $\boldsymbol{t}$ for use. Party $\boldsymbol{t}$ wants to know what the data conditions are under which the task model performs well and under which conditions it should not be used. Party $\boldsymbol{s}$ runs drift forensics and provides party $\boldsymbol{t}$ with a forensic signature, as seen in Figure 6, detailing which parameters in the data model can be changed and which should not be touched to maintain task model performance. Party $\boldsymbol{t}$ can use this information to calibrate their data processing and knows which data settings to avoid for the specific task model. As with the other drift controls, access to the raw data as well as data models is required to perform drift forensics.

## 5.3 Drift optimization

In the previous two experiments we demonstrated how raw data and a differentiable data model can be used to identify and then modularly test for unfavorable data models that should be avoided during deployment of the machine learning task model. The same mechanics can also be exploited to optimize the data itself, effectively creating a beneficial drift. In the drift optimization setting, the gradient from the task model $\boldsymbol{\Phi}_{\text{Task}}$ is propagated into the data model $\boldsymbol{\Phi}_{\text{Proc}}$ to jointly optimize both of them. In the *learned* setting, the data model parameters are jointly optimized with the task model parameters. In the *frozen* setting, only the task model parameters are optimized and the data model parameters are kept fixed[11].

### 5.3.1 Convergence and stability

In the left column (a) of Figure 7 these two scenarios are compared. The *learned* data model creates a drift that improves stability of the learning trajectory. This is indicated by the blue line which displays the validation accuracy against optimization steps for the first half of training (step 1439 corresponds to epoch 60). It exceeds that of the *frozen* data model (red line) by up to 25 percentage points in accuracy at a lower variance. For the segmentation task (bottom row Figure 7) the stabilization effect cannot be observed. This could be due to the low resolution of the problem itself as the drift optimization may not have a large effect on enhancing the solid blocks of cars in the raw data. Other evidence further suggests that inverse problems are inherently less unstable [183]. The results of the convergence and stability behavior under the different settings can also be found as a tabular summary in Table 2.

---

[11]The initialization of $\boldsymbol{\Phi}_{\text{Proc}}^{\text{para}}$ (both *frozen* and *learned*) is set to standard values which can be found in Appendix A.1 as well as in `pipeline_torch.py` of the code.

③ **Drift adjustments with $\Phi_{\text{Proc}}^{\text{para}}$**

Figure 7: Low (a) and high (b) intensity images processed by a *frozen* and a *learned* pipeline. This type of drift optimization would not be possible with processed data. The plots columns three and six display the mean of validation metrics over five cross validation runs. Column seven shows additional results on raw data for comparison. Error bars are reported as one standard deviation. Optimization step 1439 and 915 correspond to epoch 60 into training.

### 5.3.2 Helpful artifacts

In fact, the processed image from a *learned* data model with optimized drift (see *learned* column in block (a) of Figure 7 for an example) can contain visible artifacts that *aid* stability and generalization vis-a-vis the image from the *frozen* baseline data model which, arguably, looks cleaner to the human eye. A possible explanation for the improved learning trajectory could be that a varying optimized drift automatically generates samples akin to data augmentation. Such uses could further be explored in scarce data settings like fine tuning, semi-supervised or few-shot learning. Having gradient access to the data model thus offers the opportunity to optimize data generation itself for a given machine learning task. If learned data models are to be applied in real-world applications, it thus appears likely that a tradeoff has to be made between human perceived visual quality and artifacts that can be helpful to the task model.

Similar outcomes for stability and artifacts can also be observed for the reverse situation (high intensity 1.0 $x_{\text{RAW}}$) in the right column (b) of Figure 7.

### 5.3.3 Raw and data models

We demonstrated how parametrized data models can be used to optimized drift under data model constraints. Going beyond physically faithful drift controls, an interesting extension to drift optimization is training directly on raw data to optimize task model performance. While in this manuscript we are concerned with providing building blocks to emulate optical data models used in practice, training directly on raw opens up the possibility to learn purely machine-optimized optical data processing free of existing data model constraints. In the last column of Figure 7 an optimization directly on raw data is displayed for each task. Results are reported across threefold cross-validation with error bars of one standard deviation. Like in the other experiments the task model parameters are tuned as well (see task model training details in Appendix A.2). For classification, a performance similar to the *learned* setting is achieved with a more volatile optimization trajectory. For the segmentation task, the performance is not on par with either the *learned* or *frozen* setting, but it appears plausible that this gap can be substantially reduced with further finetuning. Learning directly on raw data thus appears as a promising direction for data model-free machine vision.

### 5.3.4 Use cases and limitations of drift optimization

Drift optimization can be used to squeeze out performance from a task model by creating drift that is helpful. A common use case is the adjustment of imaging pipelines, such as microscopes, that have traditionally been optimized for human end users, for example medical staff, which are increasingly being used in conjunction with automated machine learning models, for example for cell detection. By adjusting the parameters in

|  | Microscopy | Drone |
|---|---|---|
|  | Average accuracy | Average IoU |
| Learned (low) | 0.75 ± 0.09 | 0.59 ± 0.05 |
| Frozen (low) | 0.54 ± 0.21 | 0.59 ± 0.05 |
| Learned (high) | 0.78 ± 0.08 | 0.74 ± 0.04 |
| Frozen (high) | 0.67 ± 0.14 | 0.71 ± 0.05 |
| Direct raw | 0.75 ± 0.07 | 0.60 ± 0.07 |

Table 2: Tabular summary of the drift optimization results. The average accuracy and standard deviations over cross-validation runs and training steps are displayed, summarizing both the stability and converge trajectory for each setting.

the data models of existing optical laboratory infrastructure performance gains can be achieved. Due to the improved convergence and stability of the optimization trajectory it can also potentially be used in situations where compute is expensive or scarce altogther. However, these benefits do not hold across all tasks, as we saw in the case of the segmentation model, and for cases where no data models are present, such as novel or custom optical hardware, learning directly on raw data offers a promising exetension. This work targets imaging infrastructure that uses ISPs causing data drift while also allowing access to raw sensor readouts. The proposed data models enable engineers to emulate and control different data generating processes related to ISPs in a cost-effective, physically faithful manner. These models save time and money and enable new applications for data-model quality management. However, they only capture ISP-related drifts and require extensions to model other factors. The ultimate goal could be to train on RAW data, with the current pipeline serving as an interim solution until RAW data becomes more widely available.

## 6 Discussion

The main message we hope to convey in this manuscript is that black-box data models for images neither have to be the norm in machine learning research nor in engineering. Leveraging established knowledge from physical optics enables us to push the modelling goalpost further towards machine learning's core ingredient: the data. Paired with raw data, precise differentiable data models for images allow for advanced controls of dataset drift, a common and far reaching challenge across many machine learning disciplines. Applications beyond robustness validation in areas of machine learning that are also held back by black-box data, such as federated learning and formal model certifications, appear opportune, too.

Drift synthesis allows the creation of physically faithful of drift test cases. In contrast to augmentation testing, the performance drops for physically faithful test cases are less severe across the board for both uses cases in our experiments, changing the conclusions we arrive at for model selection and enabling new ways to think about generalization with targeted, data model specific deployment of task models. A plausible practical application scenario of drift synthesis for machine learning researchers and practitioners is the prospective validation of their task model to drift from different camera devices, for example microscopes across different lab sites or autonomous vehicles, without having to collect measurements from the different devices. Drift synthesis could also be interesting for other application domains that rely on data synthesis (semi- [88–90] and self-supervised learning [91, 92]) or on precise data models (aleatoric uncertainty quantification [72–82], out-of-distribution detection [34, 83–87]). While we cross-validated a substantial number of data model variations in our experiments, it should be noted that further variations, for example by reordering or adding steps, are possible. Furthermore, it should not be overlooked that dataset drift can also be caused by factors outside the ISP data model, for example the optical components of a camera. Our current data models are not yet capable of capturing factors that go beyond the ISP. Integrating work from lens manufacturing [133] to expand the reach explicit data models offers a promising next step for drift synthesis.

Drift forensics allow the precise specification of data model limitations of use for a given machine learning task model. Data models under which the task model should not be operated can be identified by gradient search and then documented. In our demonstration, the setting allowing for all parameters to be altered shows the

biggest effect on the resulting performance. Individually, changes in the black level configuration and the denoising parameters pose the greatest risk for performance of the task model at hand. Interestingly, a higher norm in the resulting RGB images does not directly translate to the most severe performance degradation of the task model. This underscores the importance of precise data models for dataset drift validation. In practice, clear specification of the limitations of use is a mandated requirement for many products that can potentially contain machine learning components, such as software as a medical device [105, 106] or autonomous vehicles [182]. Drift forensics with explicit data models can help to utilize the precision of machine learning and data engineering to satisfy such regulatory constraints. Explicit data models combined with gradient search may also be interesting to explore in areas such as formal model verification [55–71] to obtain tighter error bounds. Other constraints beyond $\ell_2$ are feasible, depending on the particular use case to be analyzed, and can be plugged into our code[12]

We also showed how differentiable data models can be used for drift optimization where the data generating process is jointly optimized with the task model parameters. It leads to improved stability of the learning trajectory on the classification task in both low and high intensity measurements. Interestingly, the processed image from a *learned* data model can contain visible artifacts that *aid* stability and generalization vis-a-vis the image from the *frozen* baseline data model which arguably looks cleaner to the human eye. In practice, the extension of the gradient connection from the task model $\mathbf{\Phi}_{\text{Task}}$ to the data model $\mathbf{\Phi}_{\text{Proc}}$ enables the extension of machine learning right into the data generating process. Thus, data generation itself can be optimized to best suit the task model at hand. Furthermore, the stabilization effect could prove useful for learning problems where training is costly and speedup precious (for example large models or large datasets). This capacity could also be exploited in other areas that deal with heterogenous training or deployment environments, such as different clients in federated learning [97–99] or domain adaptation techniques [184]. However, the above drift adjustment benefits could only be observed for the classification task, not the regression task, possibly due to the low resolution of the segmentation problem. How far we can push the gradient into the real world is an interesting future direction for data modelling. Including more parts of the data acquisition hardware into the data model and consequently the machine learning optimization pipeline appears feasible [185] and represents an important next step in aligning machine learning with real world data infrastructures.

Finally, raw data, which is already routinely used in optical industries [125–130], for representative machine learning tasks has to become more accessible to researchers to align robustness research with physically faithful data models and infrastructures. While most optical imaging devices support the extraction of raw data and this procedure is well established in industry and physics, data collection procedures for machine learning robustness research still have to catch up in order to make raw datasets and their benefits more widely available. Norms around established benchmarking datasets of processed images, such as CIFAR or ImageNet, can slow down this progress. To that end, we collected and publicly release two raw image datasets in the camera sensor state. The assumptions with respect to the practicality of the procedures we propose here are mild in our eyes. Raw subsets of data could be stored and then pulled in-code from cloud storage, as demonstrated in the code that we provide, for the purposes of drift synthesis or drift forensics. Learned data models obtained from drift adjustments could be calibrated directly on hardware such that the bandwidth requirements would not change compared to current image acquisition and transmission. Better APIs to optical hardware would allow more researchers and industries to make their raw data accessible and service a culture of data modelling that can help overcome the limitations of machine learning in the pure task model regime.

**Use of Personal Data and Human Subjects** The microscopy slides were purchased from a commercial lab vendor (J. Lieder GmbH & Co. KG, Ludwigsburg/Germany) who attained consent. The drone dataset does not directly relate to people. Instances with potential PIIs such as faces or license plates were removed. Full datasheet documentation following [177] can be found in Appendix A.4.2.

**Negative Societal Impact** Machine learning risk management, such as the drift controls, can make ML deployment possible and safer. More deployment translates to increases in automation. A net risk-benefit

---

[12]Argument `args.adv_aux_loss` in `train.py`

analysis of automation is beyond the scope of this manuscript. What we do know is that steel can be cast into ploughs and swords. We are against the use of our findings for the latter purpose.

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

# A  Appendices

## A.1  Data models details

### A.1.1  Static data model $\Phi_{\text{Proc}}^{\text{stat}}$

If not stated otherwise, writing the equation $v_{c,h,w} = a_{c,h,w} + b_{c,h,w}$ defines $v_{c,h,w}$ for all $1 \le c \le 3$, $1 \le h \le H$ and $1 \le h \le W$.

**Black level correction (BL)** removes thermal noise and readout noise generated from the camera sensor. The transformation is given by

$$\Phi_{\text{BL}} : [0,1]^{H,W} \to [0,1]^{H,W}, \boldsymbol{x}_{\text{RAW}} \mapsto \boldsymbol{v}_{\text{BL}}, \tag{10}$$

with

$$(v_{\text{BL}})_{2h+1,2w+1} = x_{2h+1,2w+1} - bl_1$$
$$(v_{\text{BL}})_{2h,2w+1} = x_{2h,2w+1} - bl_2$$
$$(v_{\text{BL}})_{2h+1,2w} = x_{2h+1,2w} - bl_3$$
$$(v_{\text{BL}})_{2h,2w} = x_{2h,2w} - bl_4,$$

By design of $\boldsymbol{bl} \in \mathbb{R}^4$, black level correction ensures that $\boldsymbol{v}_{\text{BL}}$ is again an element of $[0,1]^{H,W}$.

**Demosaicing (DM)** is applied to reconstruct the full RGB color image through interpolation. We use one out of the three demosaicing algorithms BayerBilinear ($\Phi_{\text{DM}}^{\text{Bil}}$), Menon2007 ($\Phi_{\text{DM}}^{\text{Men}}$) and Malvar2004 ($\Phi_{\text{DM}}^{\text{Mal}}$) from the Python package `color-demosaicing` and denote this transformation by the map

$$\Phi_{\text{DM}} : [0,1]^{H,W} \to [0,1]^{3,H,W}, \boldsymbol{v} \mapsto \boldsymbol{v}_{\text{DM}}. \tag{11}$$

**White balance (WB)** is applied to obtain a neutrally illuminated image. The transformation is given by

$$\Phi_{\text{WB}} : [0,1]^{3,H,W} \to [0,1]^{3,H,W}, \boldsymbol{v} \mapsto \boldsymbol{v}_{\text{WB}}, \tag{12}$$

where $\boldsymbol{wb} \in [0,1]^3$ adjusts the intensities by

$$(v_{\text{WB}})_{c,h,w} = wb_c \cdot (v_{\text{DM}})_{c,h,w}. \tag{13}$$

**Color correction (CC)** balances the saturation of the image by considering color dependencies. Let $\boldsymbol{M} \in \mathbb{R}^{3,3}$ be the color matrix. The transformation is defined by

$$\Phi_{\text{CC}} : [0,1]^{3,H,W} \to \mathbb{R}^{3,H,W}, \boldsymbol{v} \mapsto \boldsymbol{v}_{\text{CC}}, \tag{14}$$

where

$$\boldsymbol{v}_{\text{CC}} = \begin{bmatrix} (v_{\text{CC}})_{1,h,w} \\ (v_{\text{CC}})_{2,h,w} \\ (v_{\text{CC}})_{3,h,w} \end{bmatrix} = \boldsymbol{M} \begin{bmatrix} (v_{\text{WB}})_{1,h,w} \\ (v_{\text{WB}})_{2,h,w} \\ (v_{\text{WB}})_{3,h,w} \end{bmatrix}. \tag{15}$$

The entries of the resulting $\boldsymbol{v}_{\text{CC}}$ are no longer restricted to $[0,1]$.

**Sharpening (SH)** reduces the blurriness of an image. We use the two methods sharpening filter ($\Phi_{\text{SH}}^{\text{SF}}$) and unsharp masking ($\Phi_{\text{SH}}^{\text{UM}}$) that are applied after a transformation of the view $\boldsymbol{v}_{\text{CC}}$ to the $YUV$-color space. To convert the view to the $YUV$-color space we use the `skimage.color` function `rgb2yuv` ($\Phi_{YUV}$). The sharpening filter

$$SF : \mathbb{R}^{3,H,W} \to \mathbb{R}^{3,H,W}, \tag{16}$$

is defined by a channel-wise convolution

$$(SF(\boldsymbol{v}))_{c,h,w} = ((\boldsymbol{v}_c \star \boldsymbol{k})_{h,w})_c \quad with \quad \boldsymbol{k} := \begin{bmatrix} 0 & -1 & 0 \\ -1 & 5 & -1 \\ 0 & -1 & 0 \end{bmatrix} \tag{17}$$

of the view

$$\boldsymbol{v} = \boldsymbol{\Phi}_{YUV}(\boldsymbol{v}_{\mathrm{CC}}). \tag{18}$$

For unsharp masking we use the `ski.filters` function `unsharp_mask` modeled by $UM$. To formally define the sharpening we write

$$\boldsymbol{\Phi}_{\mathrm{SH}} : \mathbb{R}^{3,H,W} \to \mathbb{R}^{3,H,W}, \boldsymbol{v} \mapsto \boldsymbol{v}_{\mathrm{SH}} \tag{19}$$

where

$$\boldsymbol{v}_{\mathrm{SH}} = algo \circ \boldsymbol{\Phi}_{YUV}(\boldsymbol{v}_{\mathrm{CC}}) \quad with \quad algo \in \{SH, UM\}. \tag{20}$$

**Denoising (DN)** reduces the noise in an image that is (partly) introduced by SH and transforms the $YUV$-color space view back to the $RGB$-color space. For the latter transformation, the `skimage.color` function `yuv2rgb` ($\boldsymbol{\Phi}_{YUV}^{-1}$) is used. We apply one out of the two methods Gaussian denoising ($\boldsymbol{\Phi}_{\mathrm{DN}}^{\mathrm{GD}}$) and Median denoising ($\boldsymbol{\Phi}_{\mathrm{DN}}^{\mathrm{GD}}$). For Gaussian denoising, we apply a Gaussian filter (GF) with standard deviation of $\sigma = 0.5$ from the `scipy.ndimage` package. For median denoising we apply a median filter (MF of size 3 from the `scipy.ndimage` package. Formally, this reads as

$$\boldsymbol{\Phi}_{\mathrm{DN}} : \mathbb{R}^{3,H,W} \to \mathbb{R}^{3,H,W}, \boldsymbol{v} \mapsto \boldsymbol{v}_{\mathrm{DN}} \tag{21}$$

where

$$\boldsymbol{v}_{\mathrm{DN}} = \boldsymbol{\Phi}_{YUV}^{-1} \circ algo(\boldsymbol{v}_{\mathrm{SH}}) \quad with \quad algo \in \{GF, UM\}. \tag{22}$$

**Gamma correction (GC)** equilibrates the overall brightness of the image. First, the entries of the view $\boldsymbol{v}_{\mathrm{DN}}$ are clipped to $[0, 1]$ leading to

$$(v_{CP})_{c,h,w} = (v_{\mathrm{DN}})_{c,h,w} \mathbb{1}_{\{0 \le (v_{\mathrm{DN}})_{c,h,w} \le 1\}} + \mathbb{1}_{\{(v_{\mathrm{DN}})_{c,h,w} > 1\}}. \tag{23}$$

Second, the brightness adjusting transformation is defined by

$$\boldsymbol{\Phi}_{\mathrm{GC}} : \mathbb{R}^{3,H,W} \to [0,1]^{3,H,W}, \boldsymbol{v} \mapsto \boldsymbol{v}_{\mathrm{GC}} = (\boldsymbol{v}_{CP})^{\frac{1}{\gamma}} \tag{24}$$

for some $\gamma > 0$ applied element-wise. Note that zero-clipping is necessary for $\boldsymbol{v}_{\mathrm{GC}}$ to be well-defined.

In total, we define the composition

$$\boldsymbol{\Phi}_{Proc}^{\mathrm{stat}} : [0,1]^{H,W} \mapsto [0,1]^{3,H,W} \tag{25}$$

of the above steps

$$\boldsymbol{\Phi}_{\mathrm{Proc}}^{\mathrm{stat}} := \boldsymbol{\Phi}_{\mathrm{GC}} \circ \boldsymbol{\Phi}_{\mathrm{DN}} \circ \boldsymbol{\Phi}_{\mathrm{SH}} \circ \boldsymbol{\Phi}_{\mathrm{CC}} \circ \boldsymbol{\Phi}_{\mathrm{WB}} \circ \boldsymbol{\Phi}_{\mathrm{DM}} \circ \boldsymbol{\Phi}_{\mathrm{BL}} \tag{26}$$

and call $\boldsymbol{\Phi}_{\mathrm{Proc}}^{\mathrm{stat}}$ the *static pipeline*.

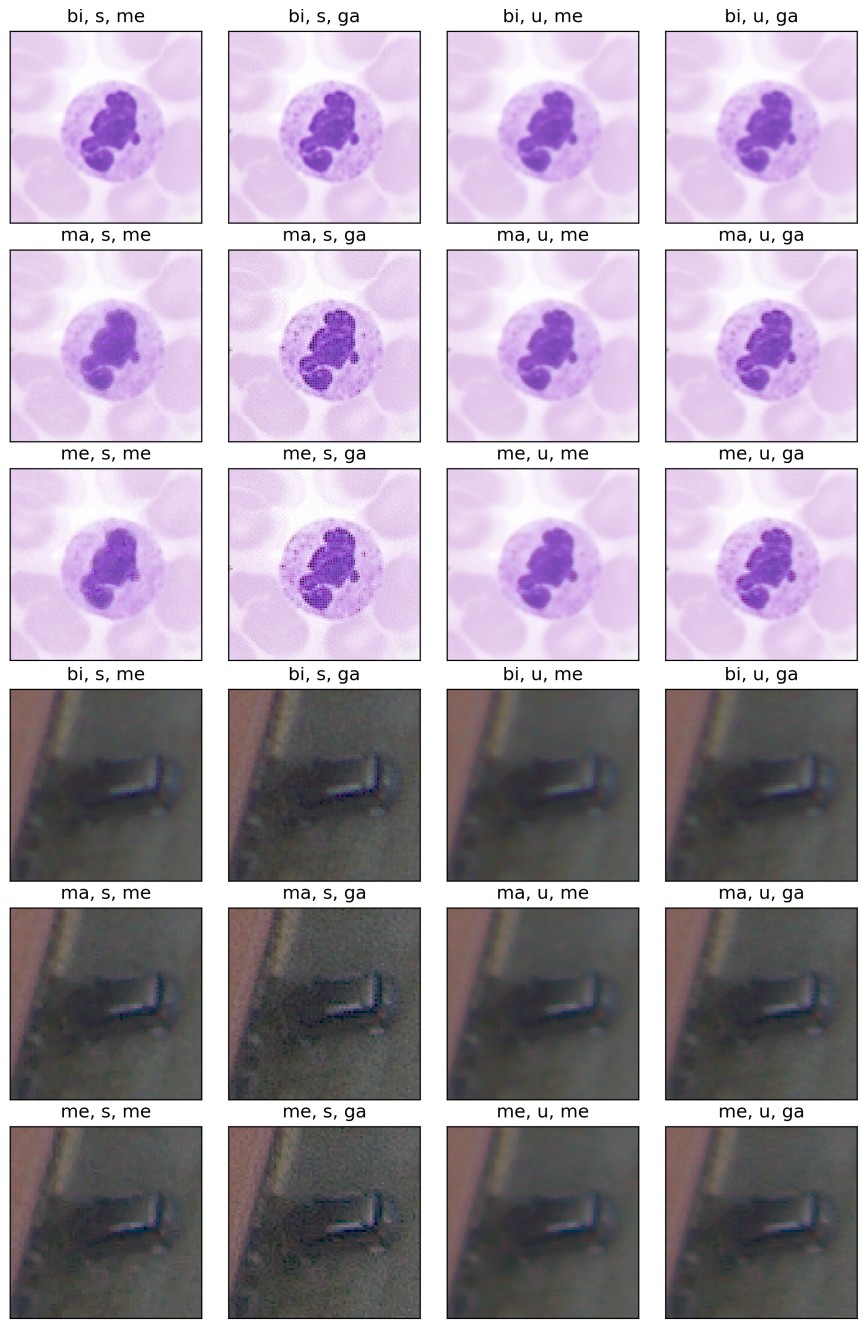

Figure 8: Samples for both datasets, Raw-Microscopy and Raw-Drone, from all twelve pipelines used in the drift synthesis experiments. The legend for abbreviations can be found in Figure 3b.

### A.1.2 Parametrized data model $\Phi_{\text{Proc}}^{\text{para}}$

**Black level correction (BL)** For the parametrized black level correction define the map

$$\Phi_{\text{BL}}^{\text{stat}} : [0,1]^{H,W} \times \mathbb{R}^4 \to \mathbb{R}^{H,W}, (\boldsymbol{x}_{\text{RAW}}, \boldsymbol{\theta}_1) \mapsto \boldsymbol{v}_{\text{BL}} = \Phi_{\text{BL}}(\boldsymbol{x}_{\text{RAW}})|_{\boldsymbol{bl}=\boldsymbol{\theta}_1}. \tag{27}$$

and set $\Theta_1 := \mathbb{R}^4$.

**Demosaicing (DM)** We first convert $\boldsymbol{v}_{\text{BL}}$ to a three channel image $[\boldsymbol{R}, \boldsymbol{G}, \boldsymbol{B}] \in \mathbb{R}^{3,H,W}$ where the entries of $\boldsymbol{R}, \boldsymbol{G}$ and $\boldsymbol{B}$ are zero except

$$R_{2h+1,2w+1} = \boldsymbol{v}_{BL_{2h+1,2w+1}}, \quad B_{2h,2w} = v_{BL_{2h,2w}},$$
$$G_{2h+1,2w} = v_{BL_{2h+1,2w}}, \quad G_{2h,2w+1} = v_{BL_{2h,2w+1}}.$$

To parametrize $\Phi_{\text{DM}}^{\text{Bil}}$ define the map

$$\Phi_{\text{DM}}^{\text{para}} : [0,1]^{H,W} \times \mathbb{R}^{3,3,3} \to \mathbb{R}^{3,H,W}, (\boldsymbol{v}_{\text{BL}}, \boldsymbol{\theta}_2) \mapsto \boldsymbol{v}_{\text{DM}} \tag{28}$$

with $\boldsymbol{\theta}_2 = [\boldsymbol{k}_1, \boldsymbol{k}_2, \boldsymbol{k}_3]$, where the kernels $\boldsymbol{k}_1, \boldsymbol{k}_2, \boldsymbol{k}_3 \in \mathbb{R}^{3,3}$ are separately applied to each color channel resuling in

$$\boldsymbol{v}_{DM_{1,h,w}} = (R \star \boldsymbol{k}_1)_{h,w}$$
$$\boldsymbol{v}_{DM_{2,h,w}} = (G \star \boldsymbol{k}_2)_{h,w}$$
$$\boldsymbol{v}_{DM_{3,h,w}} = (B \star \boldsymbol{k}_3)_{h,w}.$$

The source code of BayerBilinear shows that the parameter choice

$$\boldsymbol{k}_1 = \boldsymbol{k}_3 = \begin{bmatrix} 0 & 0.25 & 0 \\ 0.25 & 1 & 0.25 \\ 0 & 0.25 & 0 \end{bmatrix} \quad and \quad \boldsymbol{k}_2 = \begin{bmatrix} 0.25 & 0.5 & 0.25 \\ 0.5 & 1 & 0.5 \\ 0.25 & 0.5 & 0.25 \end{bmatrix} \tag{29}$$

leads to

$$\Phi_{\text{DM}}^{\text{Bil}} = \Phi_{\text{DM}}^{\text{para}}(\cdot, \boldsymbol{\theta}_2). \tag{30}$$

Towards the definition of the parameter space set $\Theta_2 := \mathbb{R}^{3,3,3} \times \Theta_1$.

**White balance (WB)** For the parametrized white balance define the map

$$\Phi_{\text{WB}}^{\text{para}} : \mathbb{R}^{3,H,W} \times \mathbb{R}^3 \to \mathbb{R}^{3,H,W}, (\boldsymbol{v}_{\text{DM}}, \boldsymbol{\theta}_3) \mapsto \boldsymbol{v}_{\text{WB}} = \Phi_{\text{WB}}(\boldsymbol{v}_{\text{DM}})|_{\boldsymbol{wb}=\boldsymbol{\theta}_3} \tag{31}$$

and set $\Theta_3 := \mathbb{R}^3 \times \Theta_2$.

**Color correction (CC)** For the parametrized color correction define the map

$$\Phi_{\text{CC}}^{\text{para}} : \mathbb{R}^{3,H,W} \times \mathbb{R}^{3,3} \to \mathbb{R}^{3,H,W}, (\boldsymbol{v}_{\text{WB}}, \boldsymbol{\theta}_4) \mapsto \boldsymbol{v}_{\text{CC}} = \Phi_{\text{CC}}(v_{\text{WB}})|_{\boldsymbol{M}=\boldsymbol{\theta}_4} \tag{32}$$

and set $\Theta_4 := \mathbb{R}^{3,3} \times \Theta_3$

**Sharpening (SH)** We parametrize the sharpening filter configuration of the static pipeline, by using the entries of $\boldsymbol{k} \in \mathbb{R}^{3,3}$ defined in (17) as parameters leading to

$$\Phi_{\text{SH}}^{\text{para}} : \mathbb{R}^{3,H,W} \times \mathbb{R}^{3,3} \to \mathbb{R}^{3,H,W}, (\boldsymbol{v}_{\text{CC}}, \boldsymbol{\theta}_5) \mapsto \boldsymbol{v}_{\text{SH}} = \Phi_{\text{SH}}(v_{\text{CC}})|_{\boldsymbol{k}=\boldsymbol{\theta}_5} \tag{33}$$

and $\Theta_5 := \mathbb{R}^{3,3} \times \theta_4$.

**Denoising (DN)** We parametrize the configuration where the Gaussian denoising method is applied. Applying the Gaussian filter from `scipy.ndimage` with $\sigma = 0.5$ is equivalent to a convolution of the view in the $YUV$-color space with a specific $\boldsymbol{k}_{gauss} \in \mathbb{R}^{5,5}$. For the specific values of $\boldsymbol{k}_{gauss}$ see `K_BLUR` at the code of the parametrized pipeline. Therefore, to parametrize DN we define the map

$$\Phi_{\text{DN}}^{\text{para}} : \mathbb{R}^{3,H,W} \times \mathbb{R}^{5,5} \to \mathbb{R}^{3,H,W}, (\boldsymbol{v}_{\text{SH}}, \boldsymbol{\theta}_6) \mapsto \boldsymbol{v}_{\text{DN}} = \Phi_{\text{DN}}(v_{\text{SH}})|_{\boldsymbol{k}_{gauss}=\boldsymbol{\theta}_6} \tag{34}$$

and set $\Theta_6 := \mathbb{R}^{5,5} \times \Theta_5$

**Gamma correction (GC)** Define the parametrized gamma correction by

$$\mathbf{\Phi}_{\mathrm{GC}}^{\mathrm{para}} : \mathbb{R}^{3,H,W} \times \mathbb{R} \to [0,1]^{3,H,W}, (\boldsymbol{v}_{\mathrm{DN}}, \boldsymbol{\theta_7}) \mapsto \boldsymbol{v} = \boldsymbol{v}_{\mathrm{GC}} = \mathbf{\Phi}_{\mathrm{GC}}(\boldsymbol{v}_{\mathrm{DN}})|_{\gamma=\boldsymbol{\theta_7}}. \tag{35}$$

The following values were used to initialize $\mathbf{\Phi}_{\mathrm{Proc}}^{\mathrm{para}}$ (both "Frozen" and "Learned") in experiment Section 5.3:

```python
class ParametrizedProcessing(nn.Module):
    """Differentiable processing pipeline via torch transformations

    Args:
        camera_parameters (tuple(list), optional): applies given camera parameters in
        processing
        track_stages (bool, optional): whether or not to retain intermediary steps in
        processing
        batch_norm_output (bool, optional): adds a BatchNorm layer to the end of the
        processing
    """

    def __init__(self, camera_parameters=None, track_stages=False, batch_norm_output=True):
        super().__init__()
        self.stages = None
        self.buffer = None
        self.track_stages = track_stages

        if camera_parameters is None:
            camera_parameters = DEFAULT_CAMERA_PARAMS

        black_level, white_balance, colour_matrix = camera_parameters

        self.black_level = nn.Parameter(torch.as_tensor(black_level))
        self.white_balance = nn.Parameter(torch.as_tensor(white_balance).reshape(1, 3))
        self.colour_correction = nn.Parameter(torch.as_tensor(colour_matrix).reshape(3, 3))

        self.gamma_correct = nn.Parameter(torch.Tensor([2.2]))

        self.debayer = Debayer()

        self.sharpening_filter = nn.Conv2d(1, 1, kernel_size=3, padding=1, bias=False)
        self.sharpening_filter.weight.data[0][0] = K_SHARP.clone()

        self.gaussian_blur = nn.Conv2d(1, 1, kernel_size=5, padding=2, padding_mode='reflect
    ', bias=False)
        self.gaussian_blur.weight.data[0][0] = K_BLUR.clone()

        self.batch_norm = nn.BatchNorm2d(3, affine=False) if batch_norm_output else None

        self.register_buffer('M_RGB_2_YUV', M_RGB_2_YUV.clone())
        self.register_buffer('M_YUV_2_RGB', M_YUV_2_RGB.clone())

        self.additive_layer = None
```

where

```python
K_G = torch.Tensor([[0, 1, 0],
                    [1, 4, 1],
                    [0, 1, 0]]) / 4

K_RB = torch.Tensor([[1, 2, 1],
                     [2, 4, 2],
                     [1, 2, 1]]) / 4

M_RGB_2_YUV = torch.Tensor([[0.299, 0.587, 0.114],
                            [-0.14714119, -0.28886916, 0.43601035],
                            [0.61497538, -0.51496512, -0.10001026]])
M_YUV_2_RGB = torch.Tensor([[1.0000000000e+00, -4.1827794561e-09, 1.1398830414e+00],
                            [1.0000000000e+00, -3.9464232326e-01, -5.8062183857e-01],
                            [1.0000000000e+00, 2.0320618153e+00, -1.2232658220e-09]])

K_BLUR = torch.Tensor([[6.9625e-08, 2.8089e-05, 2.0755e-04, 2.8089e-05, 6.9625e-08],
                       [2.8089e-05, 1.1332e-02, 8.3731e-02, 1.1332e-02, 2.8089e-05],
                       [2.0755e-04, 8.3731e-02, 6.1869e-01, 8.3731e-02, 2.0755e-04],
                       [2.8089e-05, 1.1332e-02, 8.3731e-02, 1.1332e-02, 2.8089e-05],
                       [6.9625e-08, 2.8089e-05, 2.0755e-04, 2.8089e-05, 6.9625e-08]])
K_SHARP = torch.Tensor([[0, -1, 0],
                        [-1, 5, -1],
                        [0, -1, 0]])
DEFAULT_CAMERA_PARAMS = (
    [0., 0., 0., 0.],
    [1., 1., 1.],
    [1., 0., 0., 0., 1., 0., 0., 0., 1.],
)
```

Note that the camera parameters are camera, and conversely in our case dataset, dependent and defined in the dataset classes.

### A.2 Description of the task models $\Phi_{\text{Task}}$

|  | **Classification** | **Segmentation** |
|---|---|---|
| $\Phi_{\text{Task}}$ | ResNet18 based on [180]
trained with Adam [186] for 100 epochs
learning rate: $10^{-4}$
mini-batch size: 128 | U-Net++ based on [181]
trained with Adam for 100 epochs
learning rate: $7.5 \cdot 10^{-5}$
mini-batch size: 12 |

Table 3: Summary of the training procedure for both task models.

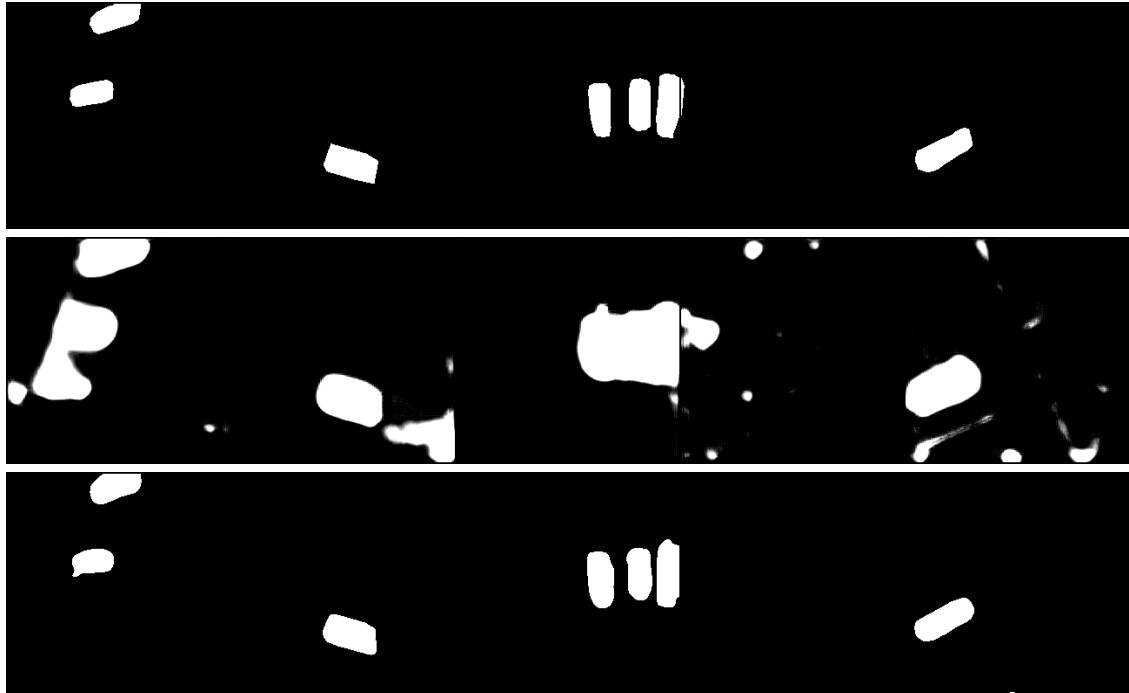

Table 4: A set of random test samples for the segmentation task under learned processing. Top row: Targets, middle row: predictions of the task model after the first epoch, last row: predictions of the task model after the last epoch.

**ResNet18** This model is designed to classify images from ImageNet [187] and has therefore an output dimension of 1000. In order to use the model to classify images from Raw-Microscopy, we changed the output dimension of the fully-connected layer to nine. The model was trained for 100 epochs using pre-trained ResNet features. Hyperparameters were kept constant across all runs to isolate the effect of varying image processing pipelines. For implementation the code provided at `https://pytorch.org/hub/pytorch_vision_resnet/` was used. The model consists of 34 layers with approximately 11.2 million trainable parameters. The storage size of the model is 44.725 MB.

**U-Net++** The model was trained for 100 epochs using pretrained ResNet features as the encoder of the U-Net++. Hyperparameters were kept constant across all runs to isolate the effect of varying image processing pipelines. For implementation we used the code provided at `https://github.com/qubvel/segmentation_models.pytorch`. The model has approximately 26.1 million trainable parameters. The storage size of the model is 104.315 MB.

**Raw training** In the drift optimization experiments of Section 5.3 the raw data is demosaiced using `class RawToRGB(nn.Module)` from `/processing/pipeline_torch.py` in the data model code. Then task models are tuned to raw data under the same regimes described above.

For a summary of the training procedure see Table 3.

## A.3 Additional results

### A.3.1 Drift synthesis

| Rank | Microscopy-ISP | | Microscopy-CC | | Drone-ISP | | Drone-CC | |
|---|---|---|---|---|---|---|---|---|
| | Train pipeline | Avg. score | Train pipeline | Avg. score | Train pipeline | Avg. score | Train pipeline | Avg. score |
| 1 | ma,s,me | 0.83 | bi,u,me | 0.63 | ma,u,ga | 0.68 | ma,s,ga | 0.60 |
| 2 | ma,u,me | 0.83 | me,s,me | 0.63 | bi,s,ga | 0.68 | bi,s,ga | 0.57 |
| 3 | ma,u,ga | 0.82 | bi,u,ga | 0.62 | bi,s,me | 0.67 | me,s,ga | 0.57 |
| 4 | bi,s,me | 0.81 | ma,s,me | 0.62 | ma,s,me | 0.67 | ma,s,me | 0.55 |
| 5 | bi,u,me | 0.81 | me,u,me | 0.62 | me,u,ga | 0.67 | me,s,me | 0.55 |
| 6 | me,s,me | 0.81 | ma,s,ga | 0.62 | me,u,me | 0.67 | ma,u,ga | 0.55 |
| 7 | bi,s,ga | 0.81 | ma,u,me | 0.61 | ma,u,me | 0.66 | bi,s,me | 0.54 |
| 8 | me,s,ga | 0.80 | me,s,ga | 0.60 | ma,s,ga | 0.66 | ma,u,me | 0.54 |
| 9 | me,u,me | 0.80 | bi,s,me | 0.59 | bi,u,me | 0.65 | me,u,me | 0.53 |
| 10 | ma,s,ga | 0.80 | ma,u,ga | 0.59 | me,s,me | 0.65 | me,u,ga | 0.51 |
| 11 | bi,u,ga | 0.79 | bi,s,ga | 0.58 | me,s,ga | 0.64 | bi,u,me | 0.48 |
| 12 | me,u,ga | 0.79 | me,u,ga | 0.58 | bi,u,ga | 0.61 | bi,u,ga | 0.46 |

Table 5: Rankings of task models from Section 5.1 trained on different data models (columns 2, 4, 6, 8) according to their average accuracy or IoU (columns 3, 5, 7, 9) across all test pipelines respective corruptions. ISP corresponds to drift synthesis with physically faithful data models, CC corresponds to common corruptions.

| Rank | Microscopy-ISP | | | | | | | | | | | |
|---|---|---|---|---|---|---|---|---|---|---|---|---|
| | bi,s,me | bi,s,ga | bi,u,me | bi,u,ga | ma,s,me | ma,s,ga | ma,u,me | ma,u,ga | me,s,me | me,s,ga | me,u,me | me,u,ga |
| 1 | ma,u,me | ma,u,me | ma,u,ga | ma,u,ga | ma,s,me | ma,u,ga | ma,u,ga | ma,u,ga | ma,u,me | me,s,ga | ma,u,ga | ma,u,ga |
| 2 | ma,u,ga | ma,u,ga | bi,s,ga | bi,s,ga | bi,s,me | me,s,ga | ma,s,me | ma,u,me | ma,s,me | ma,u,ga | ma,u,me | ma,u,me |
| 3 | bi,s,ga | bi,s,ga | ma,s,me | ma,s,me | bi,u,ga | ma,s,ga | ma,u,me | ma,s,me | bi,s,ga | ma,s,ga | ma,s,me | ma,s,me |
| 4 | ma,s,me | ma,s,me | ma,u,me | ma,u,me | ma,u,me | ma,s,me | bi,s,ga | me,u,me | me,s,ga | me,u,ga | me,u,me | me,u,me |
| 5 | bi,s,me | bi,u,me | me,u,me | me,u,me | bi,u,me | ma,u,me | me,u,me | ma,s,ga | bi,u,me | me,s,me | bi,s,ga | bi,s,ga |
| 6 | bi,u,me | me,u,me | bi,u,me | bi,u,me | ma,u,ga | me,s,me | me,s,me | bi,s,ga | ma,u,ga | ma,u,me | me,u,ga | me,u,ga |
| 7 | me,s,me | bi,s,me | bi,s,me | me,s,me | me,s,me | me,u,me | me,s,me | me,s,ga | me,u,me | ma,s,me | me,s,me | me,s,me |
| 8 | me,s,ga | me,s,me | me,s,me | bi,u,ga | bi,s,ga | bi,u,me | ma,s,ga | me,s,me | me,u,me | me,u,me | bi,s,me | bi,s,me |
| 9 | me,u,me | me,s,ga | bi,u,ga | bi,s,me | me,s,ga | me,u,ga | bi,u,me | bi,u,me | bi,s,me | bi,s,me | me,s,ga | me,s,ga |
| 10 | ma,s,ga | ma,s,ga | ma,s,ga | ma,s,ga | ma,s,ga | bi,s,me | bi,s,me | ma,s,ga | bi,s,ga | ma,s,ga | ma,s,ga | ma,s,ga |
| 11 | bi,u,ga | me,u,ga | me,u,ga | me,s,ga | me,u,ga | bi,s,ga | me,u,ga | me,u,ga | me,u,ga | bi,u,me | bi,u,me | bi,u,me |
| 12 | me,u,ga | bi,u,ga | me,s,ga | me,u,ga | me,u,me | bi,u,ga | bi,u,ga | bi,u,ga | bi,u,ga | bi,u,ga | bi,u,ga | bi,u,ga |

Table 6: Ranking of task models from Section 5.1 trained under different train pipelines (rows) for each individual test pipeline (columns 2 - 13).

| Rank | identity | gauss noise | shot | impulse | speckle | Microscopy-CC gauss blur | zoom | contrast | brightness | saturate | elastic |
|---|---|---|---|---|---|---|---|---|---|---|---|
| 1 | ma,u,me | ma,u,me | bi,u,me | bi,u,me | ma,s,ga | bi,s,ga | bi,s,ga | bi,s,ga | me,s,me | ma,s,me | bi,s,ga |
| 2 | ma,u,ga | ma,s,ga | ma,s,ga | me,u,me | bi,u,me | ma,u,me | ma,u,ga | bi,u,ga | ma,s,me | me,u,me | ma,u,ga |
| 3 | bi,s,ga | me,u,me | me,s,me | bi,u,ga | me,s,me | ma,u,ga | ma,s,me | me,u,ga | bi,u,ga | me,s,me | ma,u,me |
| 4 | me,s,me | me,s,ga | ma,u,me | me,s,me | me,u,me | bi,u,me | ma,u,me | ma,s,me | ma,s,ga | bi,u,ga | ma,s,me |
| 5 | ma,s,me | bi,u,me | me,s,ga | ma,s,me | bi,u,ga | me,u,me | bi,u,me | ma,u,me | bi,s,me | bi,s,ga | me,u,me |
| 6 | me,u,me | ma,u,ga | me,u,me | ma,u,me | ma,s,me | ma,s,me | me,s,me | bi,s,me | bi,u,me | bi,u,me | me,s,ga |
| 7 | me,s,ga | me,s,me | bi,s,me | ma,u,ga | ma,u,me | me,s,ga | bi,u,ga | bi,u,me | me,s,ga | ma,u,ga | me,s,me |
| 8 | bi,u,me | bi,s,me | bi,u,ga | me,s,ga | me,s,ga | ma,s,ga | me,u,ga | me,s,me | ma,u,ga | ma,s,ga | bi,u,ga |
| 9 | bi,u,ga | ma,s,me | ma,s,me | me,u,ga | bi,s,me | me,s,me | me,u,me | ma,s,ga | me,u,ga | bi,s,me | bi,u,me |
| 10 | ma,s,ga | bi,u,ga | ma,u,ga | ma,s,ga | ma,u,ga | bi,u,ga | me,s,ga | ma,u,ga | bi,s,ga | me,s,ga | ma,s,ga |
| 11 | bi,s,me | bi,s,ga | bi,s,ga | bi,s,me | me,u,ga | bi,s,me | ma,s,ga | me,u,me | me,u,me | me,u,ga | me,u,ga |
| 12 | me,u,ga | me,u,ga | me,u,ga | bi,s,ga | bi,s,ga | me,u,ga | bi,s,me | me,s,ga | ma,u,me | ma,u,me | bi,s,me |

Table 7: Ranking of task models from Section 5.1 trained under different train pipelines (rows) for each individual test corruptions (columns 2 - 12).

| Rank | bi,s,me | bi,s,ga | bi,u,me | bi,u,ga | ma,s,me | Drone-ISP ma,s,ga | ma,u,me | ma,u,ga | me,s,me | me,s,ga | me,u,me | me,u,ga |
|---|---|---|---|---|---|---|---|---|---|---|---|---|
| 1 | bi,s,me | bi,s,ga | bi,u,me | bi,u,me | ma,u,ga | ma,s,ga | ma,u,ga | ma,u,ga | ma,s,me | ma,s,ga | ma,u,ga | ma,u,ga |
| 2 | bi,u,me | bi,s,me | bi,s,me | bi,s,me | ma,s,me | me,s,ga | me,u,me | me,u,me | me,u,me | ma,s,ga | me,u,me | me,u,ga |
| 3 | ma,u,ga | ma,u,ga | bi,u,ga | bi,u,ga | bi,s,ga | ma,s,me | ma,u,me | ma,u,me | ma,s,me | ma,s,me | ma,s,me | me,u,me |
| 4 | bi,s,ga | ma,s,me | ma,u,ga | ma,u,ga | me,u,ga | me,s,me | bi,s,me | bi,s,me | bi,s,ga | me,s,me | me,u,ga | ma,s,me |
| 5 | me,u,me | me,u,ga | me,u,me | me,u,me | ma,s,ga | bi,s,ga | ma,s,me | ma,s,me | me,u,ga | bi,s,ga | ma,u,me | ma,u,me |
| 6 | bi,u,ga | ma,s,ga | bi,s,ga | bi,s,ga | ma,u,me | ma,u,ga | bi,s,ga | bi,s,ga | me,s,me | ma,u,ga | bi,s,me | bi,s,me |
| 7 | ma,s,me | ma,u,me | ma,u,me | ma,u,me | me,u,me | me,u,me | me,u,ga | me,u,ga | me,u,ga | me,s,ga | me,u,ga | bi,s,ga |
| 8 | me,u,ga | me,s,ga | ma,s,me | ma,s,me | me,s,me | me,u,ga | bi,u,me | bi,u,me | me,u,me | me,u,me | bi,s,ga | bi,u,me |
| 9 | ma,u,me | me,u,me | me,u,ga | me,u,ga | bi,s,me | ma,u,me | bi,u,me | ma,s,ga | ma,u,me | ma,u,me | me,s,me | ma,s,ga |
| 10 | me,s,me | me,s,me | me,s,me | me,s,me | me,s,ga | bi,s,me | ma,s,ga | me,s,me | bi,s,me | bi,s,me | bi,u,ga | me,s,me |
| 11 | ma,s,ga | bi,u,me | me,s,ga | ma,s,ga | bi,u,me | bi,u,me | me,s,me | bi,u,ga | bi,u,me | bi,u,me | ma,s,ga | bi,u,ga |
| 12 | me,s,ga | bi,u,ga | ma,s,ga | me,s,ga | bi,u,ga | bi,u,ga | me,s,ga | me,s,ga | me,s,ga | bi,u,ga | me,s,ga | me,s,ga |

Table 8: Ranking of task models from Section 5.1 trained under different train pipelines (rows) for each individual test pipeline (columns 2 - 13).

| Rank | identity | gauss noise | shot | impulse | speckle | Drone-CC gauss blur | zoom | contrast | brightness | saturate | elastic |
|---|---|---|---|---|---|---|---|---|---|---|---|
| 1 | ma,s,ga | ma,s,ga | ma,s,ga | ma,s,ga | ma,s,ga | ma,s,ga | bi,s,me | bi,s,ga | bi,s,ga | ma,s,ga | ma,s,ga |
| 2 | bi,s,ga | me,s,ga | me,s,ga | me,s,ga | me,s,ga | bi,s,ga | ma,s,ga | ma,s,ga | ma,s,ga | ma,s,me | ma,u,ga |
| 3 | me,s,ga | bi,s,ga | bi,u,ga | me,s,me | bi,s,ga | ma,s,me | bi,s,ga | me,s,me | ma,s,me | ma,u,ga | ma,s,me |
| 4 | ma,s,me | me,s,me | ma,s,me | bi,s,ga | ma,s,me | ma,u,ga | me,s,ga | ma,s,me | me,s,me | me,u,ga | bi,s,ga |
| 5 | ma,u,ga | ma,u,ga | me,s,me | ma,u,ga | me,s,me | bi,u,me | ma,u,me | bi,s,me | ma,u,me | me,s,ga | bi,s,me |
| 6 | bi,s,me | ma,u,me | ma,u,ga | ma,u,me | ma,u,ga | bi,s,me | me,s,me | ma,u,me | ma,u,ga | bi,s,ga | bi,u,me |
| 7 | me,u,ga | me,u,me | ma,u,me | me,u,me | bi,s,me | me,s,ga | ma,s,me | ma,u,ga | me,u,me | bi,s,me | me,s,ga |
| 8 | bi,u,me | ma,s,me | bi,s,me | ma,s,me | ma,u,me | ma,u,me | bi,u,me | me,s,ga | bi,s,me | me,s,me | me,u,me |
| 9 | ma,u,me | bi,s,me | me,u,me | bi,s,me | me,u,me | me,u,me | me,u,me | bi,u,me | me,u,ga | me,u,me | me,u,ga |
| 10 | me,u,me | me,u,ga | me,u,ga | me,u,ga | me,u,ga | me,s,me | bi,u,ga | bi,u,ga | me,s,ga | bi,u,me | me,s,me |
| 11 | me,s,me | bi,u,me | bi,u,me | bi,u,me | bi,u,me | me,u,ga | ma,u,ga | me,u,ga | bi,u,me | ma,u,me | ma,u,me |
| 12 | bi,u,ga | bi,u,ga | bi,u,ga | bi,u,ga | bi,u,ga | bi,u,ga | me,u,ga | me,u,me | bi,u,ga | bi,u,ga | bi,u,ga |

Table 9: Ranking of task models from Section 5.1 trained under different train pipelines (rows) for each individual test corruptions (columns 2 - 12).

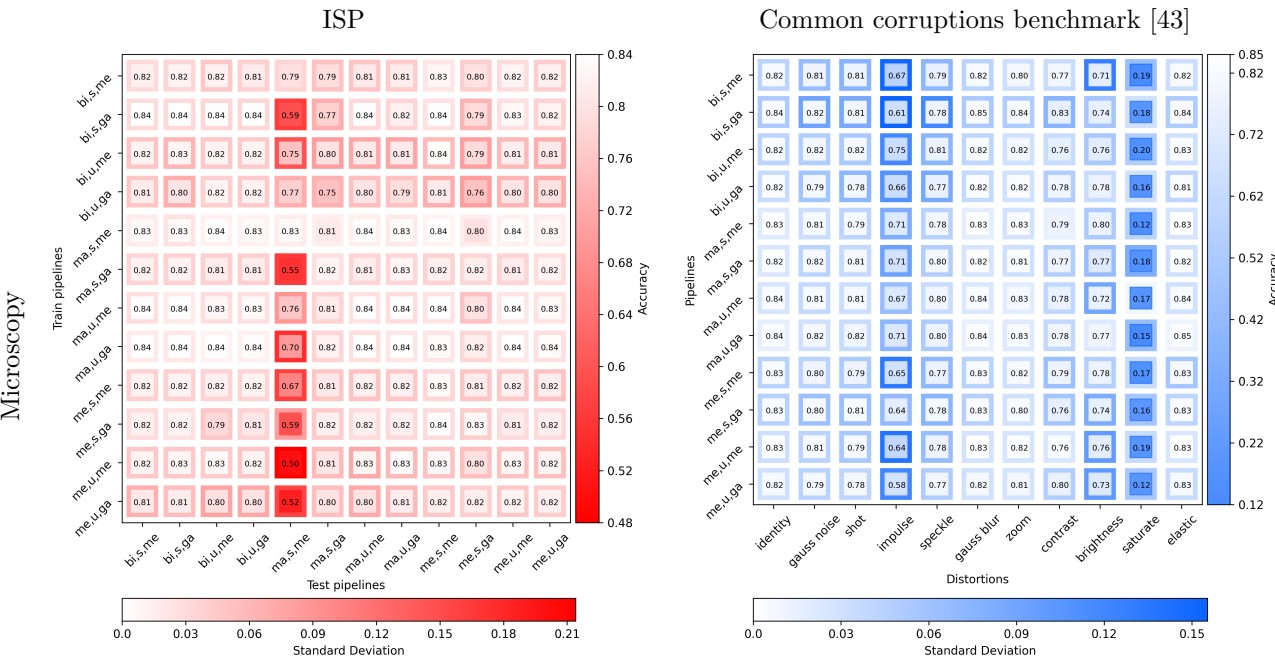

Figure 9: Experiment from Section 5.1 with weak severity (level 1) for the Common corruptions benchmark.

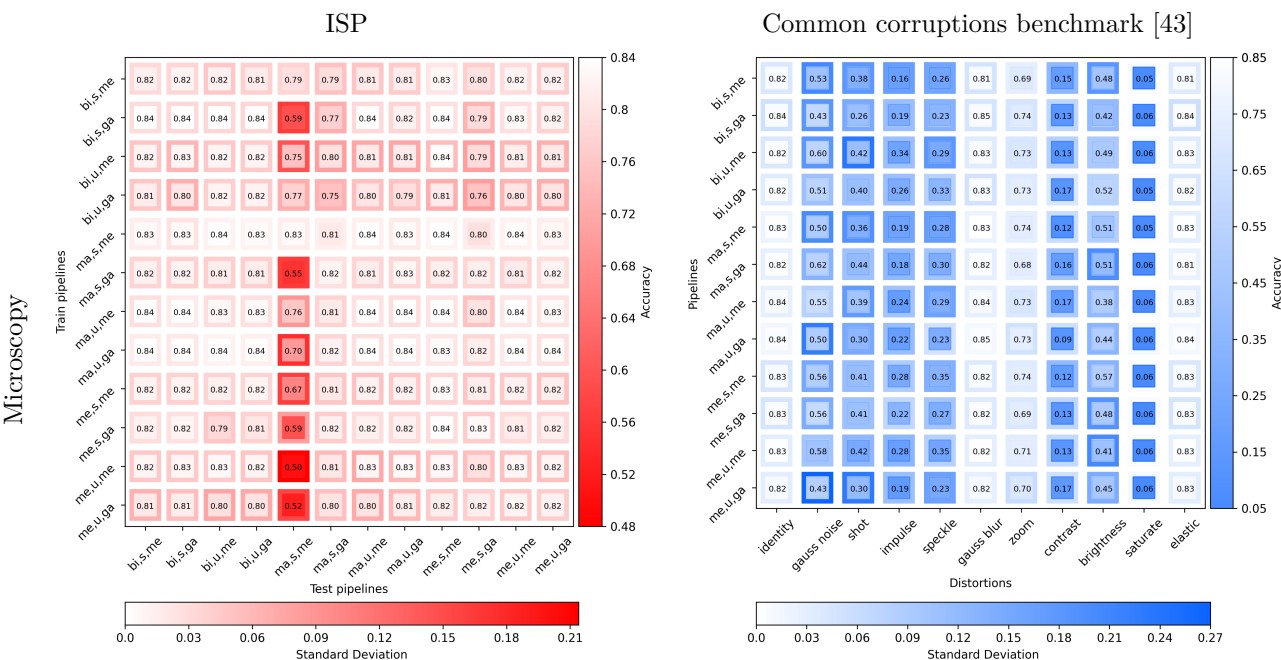

Figure 10: Experiment from Section 5.1 with strong severity (level 5) for the Common corruptions benchmark.

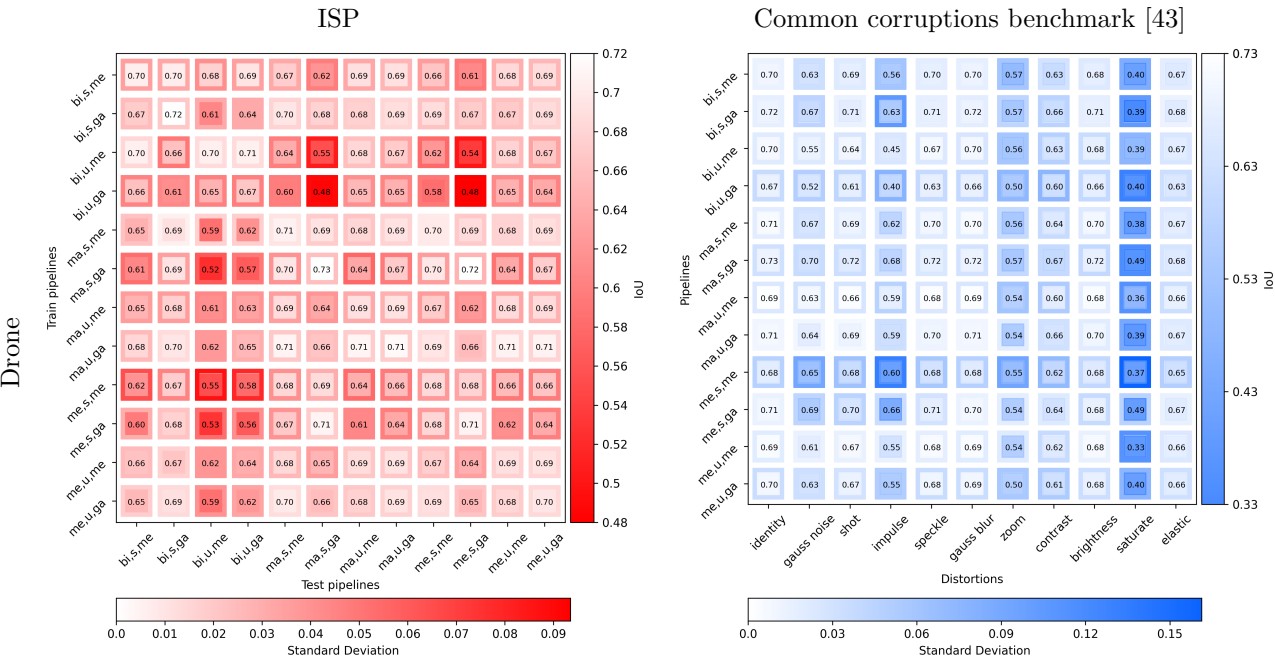

Figure 11: Experiment from Section 5.1 with weak severity (level 1) for the Common corruptions benchmark.

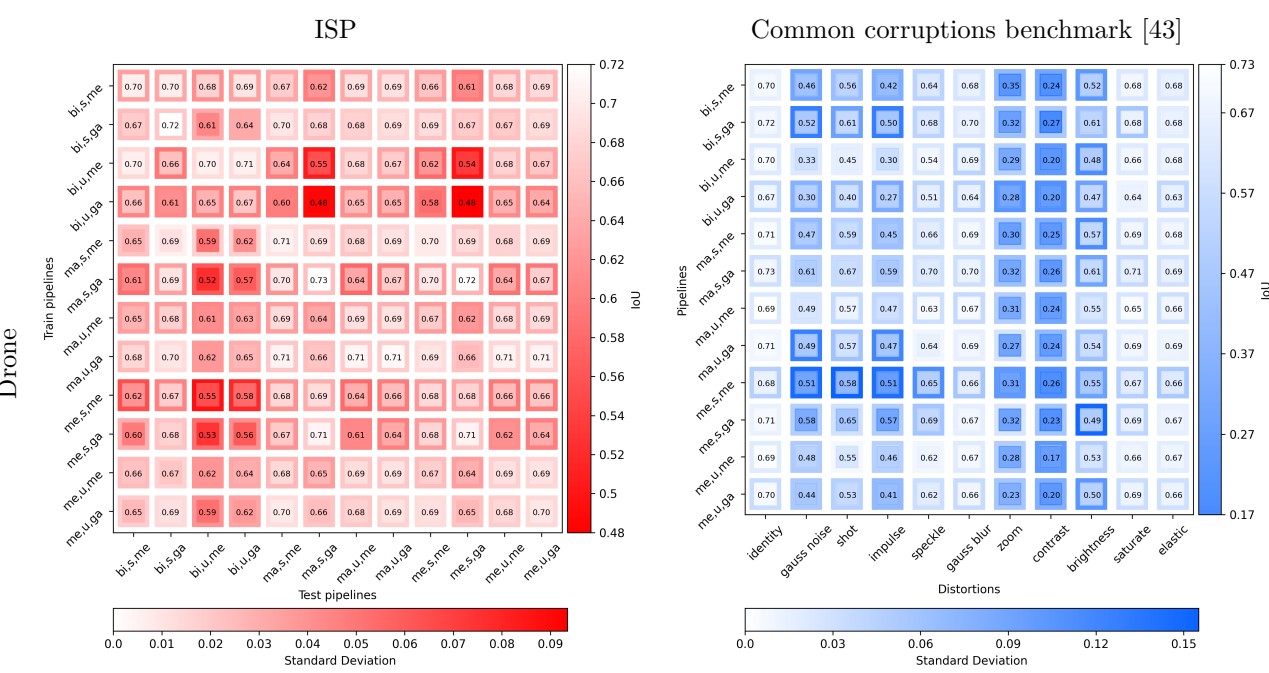

Figure 12: Experiment from Section 5.1 with strong severity (level 5) for the Common corruptions benchmark.

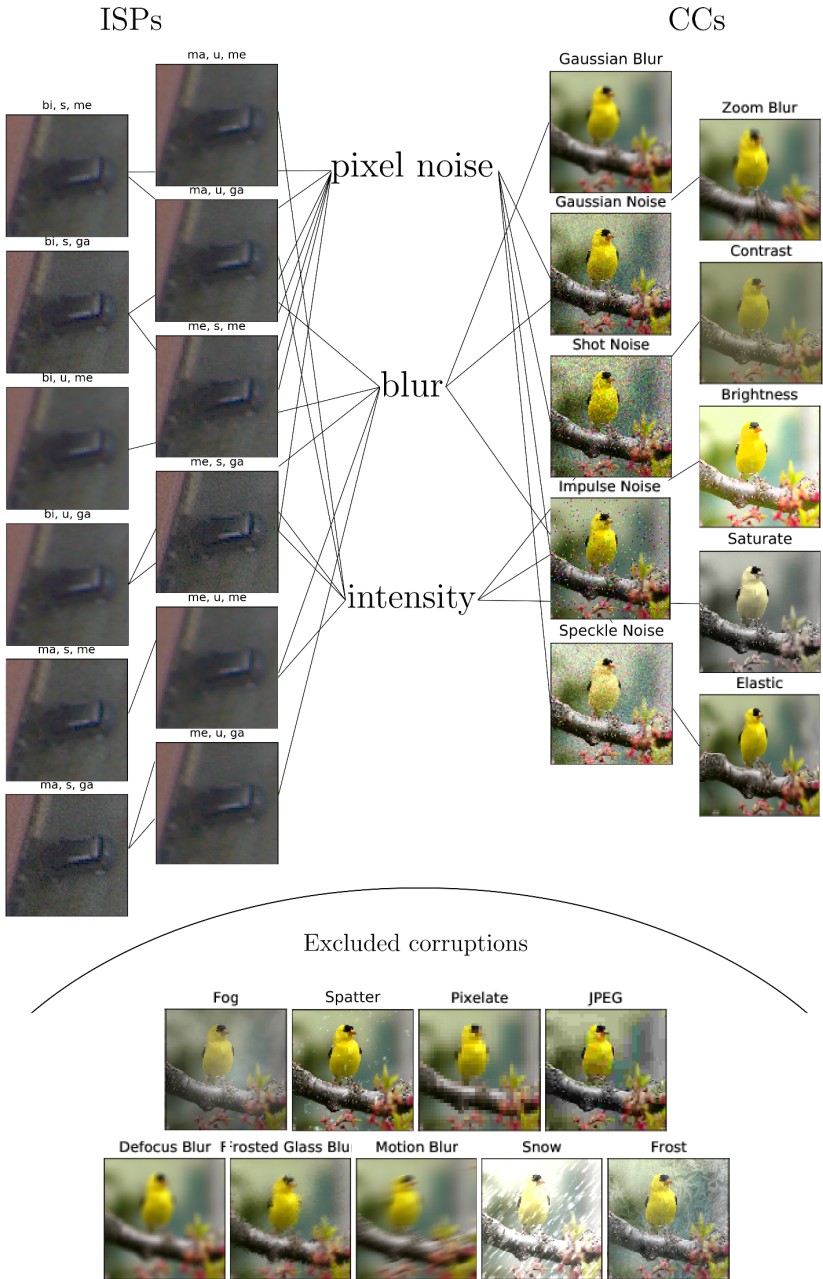

Figure 13: A comparative overview of the physically faithful data models (ISPs, top-left) and the Common Corruptions (CC, top-right) used in the the drift synthesis experiments of Section 5.1. A matching heuristic based on possible visual perception of the drift artifacts (top-middle) is provided for readers who would like to relate specific data models to specific corruptions. However, we emphasize that this is a *purely qualitative heuristic* and has no metrological basis. Since CCs are not physically faithful it is not clear how to relate them to actual variations in the optical data generating process. Finally, corruptions that were excluded from the experiments in Section 5.1 are displayed (bottom). The CC examples where stitched from the original paper [188] for authenticity.

### A.4 Datasets details

#### A.4.1 Data acquisition

In the following, core information on the two acquired datasets is provided. In Appendix A.4.2 you can also find detailed datasheets for both datasets, following the documentation good practices introduced by [177].

**Raw-Microscopy** Assessment of blood smears under a light microscope is a key diagnostic technique for many healthcare services such as cancer treatment and kidney failure as well as blood disorder detection [189]. The creation of image datasets and machine learning models on them has received wide interest in recent years [13, 190, 191]. Variations in the image processing can affect the downstream task model performance [192]. Dataset drift controls can thus help to specify the perimeter of safe application for a task model. A raw dataset was collected for that purpose. A bright-field microscope was used to image blood smear cytopathology samples. The light source is a halogen lamp equipped with a 0.55 NA condenser, and a pre-centred field diaphragm unit. Filters at 450 nm, 525 nm and 620 nm were used to acquire the blue, green and red channels respectively. The condenser is followed by a 40× objective with 0.95 NA (Olympus UPLXAPO40X). Slides can be moved via a piezo with 1 nm spatial resolution, in three directions. Focus was achieved by maximizing the variance of the pixel values[13]. Images are acquired at 16 bit, with a 2560 × 2160 pixels CMOS sensor (PCO edge 5.5). The point-spread function (PSF) was measured to be 450 nm with 100 nm nanospheres. Mechanical drift was measured at 0.4 pixels per hour. Imaging was performed on de-identified human blood smear slides (Ma190c Lieder, J. Lieder GmbH & Co. KG, Ludwigsburg/Germany). All slides were taken from healthy humans without known hematologic pathology. Imaging regions were selected to contain single leukoytes in order to allow unique labelling of image patches, and regions were cropped to 256 × 256 pixels. All images were annotated by a trained hematological cytologist using the standard scheme of normal leukocytes comprising band and segmented neutrophils, typical and atypical lymphocytes, monocytes, eosinophils and basophils [193]. To soften class imbalance, candidates for rare normal leukocyte types were preferentially imaged, and enrich rare classes. Additionally, two classes for debris and smudge cells, as well as cells of unclear morphology were included. Labelling took place for all imaged cells from a particular smear at a time, with single-cell patches shown in random order. Raw images were extracted using JetRaw Data Suite features. Blue, red and green channels are metrologically rescaled independently in intensity to simulate a standard RGB camera condition. Some pixels are discarded complementary on each channel in order to obtain a Bayer filter pattern.

Raw-Microscopy for segmentation comes with 940 raw images, twelve differently processed variants totaling 11280 images and six additional raw intensity levels totaling 5640 samples.

**Raw-Drone** Automated processing of drone data has useful applications including precision agriculture [194] or environmental protection [195]. Variation in image processing has been shown to affect task model performance [111, 115], underlining the need for drift controls. For the purposes of this study, a raw car segmentation dataset was created for the drone image modality. A DJI Mavic 2 Pro Drone was used, equipped with a Hasselblad L1D-20c camera (Sony IMX183 sensor) having 2.4 μm pixels in Bayer filter array. The lens has a focal length of 10.3 mm. The f-number was set to $N = 8$, to emulate the PSF circle diameter relative to the pixel pitch and ground sampling distance (GSD) as would be found on images from high-resolution satellites. The PSF was measured to have a circle diameter of 12.5 μm. This corresponds to a diffraction-limited system, within the uncertainty dominated by the wavelength spread of the image. Images were taken at 200 ISO, a gain of 0.528 DN/$e^-$. The 12-bit pixel values are however left-justified to 16-bits, so that the gain on the 16-bit numbers is 8.448 DN/$e^-$. The images were taken at a height of 250 m, so that the GSD is 6 cm. All images were tiled in 256 × 256 patches. Segmentation masks were created to identify cars for each patch. From this mask, classification labels were generated to detect if there is a car in the image. The dataset is constituted by 548 images for the segmentation task.

Raw-Drone for segmentation comes with 548 raw images, twelve differently processed variants totaling 6576 images and six additional raw intensity levels totaling 3288 samples.

---

[13]Figure 14 in Appendix A.4.1 provides an illustration of the imaging setup.

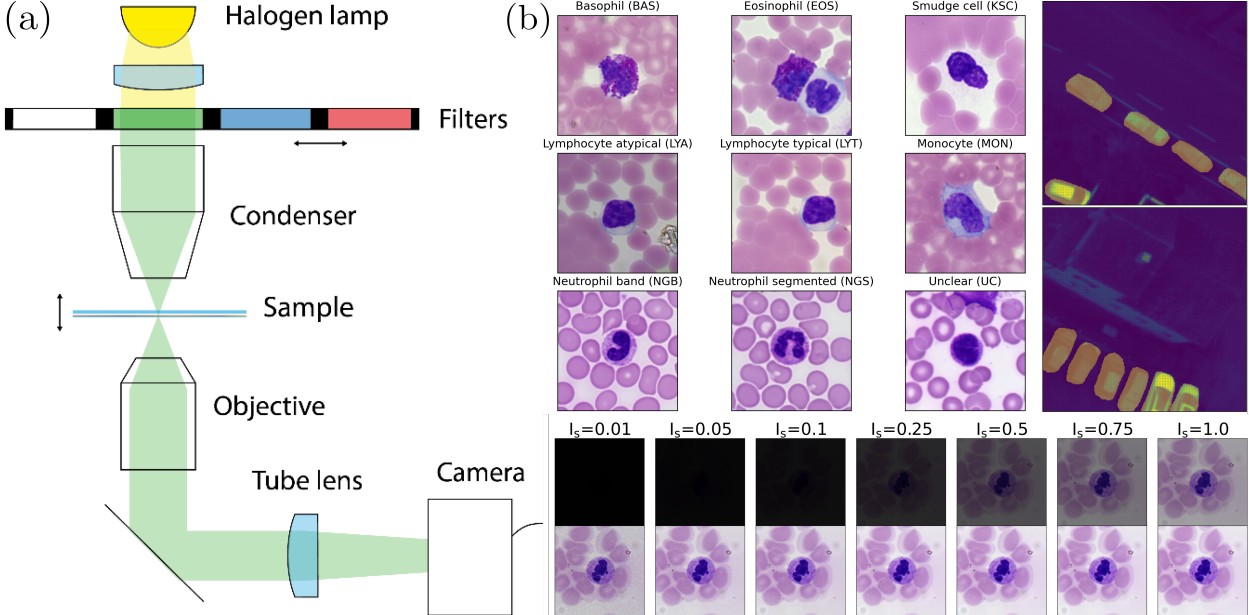

Figure 14: (a) An illustration of the imaging setup. (b) Datasets visualization. (Top-left) RGB raw microscopy classes are shown. (Top-right) Drone raw images are shown with the segmentation mask applied over it. (Bottom) Different intensity realizations are shown for the microscopy case. Images on the top are directly print out in the same scale of the original image. Images in the bottom row are normalized on their own min and max values to highlight the role of noise levels on low intensity images.

| Composition of Raw-Microscopy | |
|---|---|
| Type of instances | Image and label |
| Objects on images | White blood cells |
| Type of classes | Morphological classes |
| Number of instances | 940 |
| Number of classes | 9 |
| Image size | 256 by 256 pixels |
| Image format | .tif |
| Raw image format | Please see Section 4.1 |

| Class | Proportion in % |
|---|---|
| Basophil (BAS) | 1.91 |
| Eosinophil (EOS) | 5.74 |
| Smudge cell / debris (KSC) | 17.34 |
| atypical Lymphocyte (LYA) | 3.19 |
| typical Lymphocyte (LYT) | 24.47 |
| Monocyte (MON) | 20.32 |
| Neutrophil (band) (NGB) | 0.85 |
| Neutrophil (segmented) (NGS) | 22.98 |
| Image that could not be assigned a class (UNC) | 3.19 |

| Composition of Raw-Drone | |
|---|---|
| Type of instances | Image and mask |
| Objects on images | Landscape shots from above |
| Number of instances | 548 |
| Number of original images | 12 |
| Image size | 256 by 256 pixels |
| Mask size | 256 by 256 pixels |
| Original image size | 3648 by 5472 |
| Image format | .tif |
| Mask format | .png |
| Raw image format | .DNG |

Table 10: Summaries of the compositions of Raw-Microscopy and Raw-Drone

### A.4.2 Datasheets

We follow the datasheets documentation framework proposed in [177], using the template `https://de.overleaf.com/latex/templates/datasheet-for-dataset-template/jgqyyzyprxth` from Christian Garbin.

**Datasheet for Raw-Microscopy**

---

| Motivation |
|:----------:|

**For what purpose was the dataset created?**

With Raw-Microscopy we provide a publicly available raw image dataset in order to examine the effect of the image signal processing on the performance and the robustness of machine learning models. This dataset enables to study these effects for a supervised multiclass classification task: the classification of white blood cells (WBCs).

**Who created this dataset (e.g., which team, research group) and on behalf of which entity (e.g., company, institution, organization)?**

This dataset has been created by the Laboratory of Applied Optics of the Micro-Nanotechnology group at HEPIA/HES-SO, University of Applied Sciences of Western Switzerland. Single-cell images were annotated by a trained cytologist.

**Who funded the creation of the dataset?**

The creation of the dataset has been funded by HEPIA/HES-SO.

---

| Composition |
|:-----------:|

**What do the instances that comprise the dataset represent (e.g., documents, photos, people, countries)?**

An instance is a tuple of an image and a label. The image shows a human WBCs and the label indicates the morphological class of this cell. The following eight morphological classes appear in the dataset: Basophil (BAS), Eosinophil (EOS), Smudge cell / debris (KSC), atypical Lymphocyte (LYA), typical Lymphocyte (LYT ), Monocyte (MON), Neutrophil (band) (NGB), Neutrophil (segmented) (NGS). The nith class consists of images that could not be assigned a class (UNC) during the labeling process.

**How many instances are there in total (of each type, if appropriate)?**

The data set consists of 940 instances. For the proportion of each class in the dataset see table 11.

**Does the dataset contain all possible instances or is it a sample (not necessarily random) of instances from a larger set?**

The dataset does not contain all possible instances. It is limited to WBC classes normally present in the peripheral blood of healthy humans. In order to cope with intrinsic class imbalance in cell distribution, rare cell class candidates such as Basophils were preferentially imaged.

**What data does each instance consist of? "Raw" data (e.g., unprocessed text or images) or features?**

Each instance consists of an image of 256 by 256 pixels. The image is a raw image in .tiff format.

**Is there a label or target associated with each instance?**

Each instance is associated to a label, that indicates the morphological class of the image.

**Is any information missing from individual instances?**

No information is missing.

**Are relationships between individual instances made explicit (e.g., users' movie ratings, social network links)?**

No, relationships between individuals are not made explicit.

**Are there recommended data splits (e.g., training, development/validation, testing)?**

There are no recommended data splits. All the data splits that we used for our experiments were randomly picked.

**Are there any errors, sources of noise, or redundancies in the dataset?**

To the best of our knowledge, there are no errors in the dataset. However, a key source of variability between slides from different laboratories and processing times is stain intensity. The samples used in this work all come from the same source, hence we assume the preanalytic treatment and staining protocol to be similar. As all images were obtained on the same microscopy equipment, focus handling and illumination are identical for all samples. Image labelling was performed by one trained morphologist with experience in hematological routine diagnostics. It is known that morphology annotations are subject to inter- and intra-rater variability. However, as we limit ourselves to normal WBCs the labeling is expected to be stable.

**Is the dataset self-contained, or does it link to or otherwise rely on external resources (e.g., websites, tweets, other datasets)?**

The dataset is self-contained.

**Does the dataset contain data that might be considered confidential (e.g., data that is protected by legal privilege or by doctor-patient confidentiality, data that includes the content of individuals non-public communications)?**

The dataset consist of medical data, disclosing the morphological classes of single human WBCs. In principle, the distribution of cell types conveys information on the health state of a patient.
However, the subjects in this dataset are fully deidentified, so that the image data cannot be linked back to the healthy donors of the scanned blood smears. Furthermore, it is not disclosed which cell image was taken from which blood smear, so that no frequencies of individual cell types can be determined. Additionally, we only consider cell types present in normal blood, so that no specific hematologic pathology can be deduced from cell morphologies.

**Does the dataset contain data that, if viewed directly, might be offensive, insulting, threatening, or might otherwise cause anxiety?**

No. The dataset does not contain data with any of the above properties.

**Does the dataset relate to people?**

Yes. The dataset consist of images of human WBCs.

**Does the dataset identify any subpopulations (e.g., by age, gender)?**

The donors of the blood smears used in this dataset are fully deidentified, and no information on subpipulation composition is provided.

**Is it possible to identify individuals (i.e., one or more natural persons), either directly or indirectly (i.e., in combination with other data) from the dataset?**

No. It is not possible to identify individuals from an image of their white blood cells or visa versa.

**Does the dataset contain data that might be considered sensitive in any way (e.g., data that reveals racial or ethnic origins, sexual orientations, religious beliefs, political opinions or union memberships, or locations; financial or health data; biometric or genetic data; forms of government identification, such as social security numbers; criminal history)?**

No. While the distribution of cell types for a specific patient could reveal information about that patient's health status, isolated single-cell images of normal leukocytes do not allow for this inference.

**Any other comments?**

See table 12 for a summary of the composition of Raw-Microscopy.

| Class | Proportion in % |
|---|---|
| Basophil (BAS) | 1.91 |
| Eosinophil (EOS) | 5.74 |
| Smudge cell / debris (KSC) | 17.34 |
| atypical Lymphocyte (LYA) | 3.19 |
| typical Lymphocyte (LYT) | 24.47 |
| Monocyte (MON) | 20.32 |
| Neutrophil (band) (NGB) | 0.85 |
| Neutrophil (segmented) (NGS) | 22.98 |
| Image that could not be assigned a class (UNC) | 3.19 |

Table 11: The proportion of the classes in Raw-Microscopy.

## Collection Process

**How was the data associated with each instance acquired?**

Images of the dataset have been acquired directly from a CMOS imaging sensor. They are in a raw unprocessed format.

**What mechanisms or procedures were used to collect the data (e.g., hardware apparatus or sensor, manual human curation, software program, software API)?**

Imaging data have been obtained via a custom bright-field microscope.

**If the dataset is a sample from a larger set, what was the sampling strategy (e.g., deterministic, probabilistic with specific sampling probabilities)?**

Images have 256×256 pixel size and have been cropped from larger images. The dataset corresponds to a selection of white blood cells in the acquired large images. A sampling strategy aimed at increasing the proportion of rare classes of white blood cells has been used.

**Who was involved in the data collection process (e.g., students, crowdworkers, contractors) and how were they compensated (e.g., how much were crowdworkers paid)?**

A research assistant has been involved in the data collection process and has been compensated with a monthly salary.

**Over what timeframe was the data collected? Does this timeframe match the creation timeframe of the data associated with the instances (e.g., recent crawl of old news articles)?**

Data have been collected on a timeframe of two months, corresponding to the availability of the physical samples to image. Data have been collected on purpose for this work.

**Were any ethical review processes conducted (e.g., by an institutional review board)?**

The microscopy data was purchased from a commercial lab vendor (J. Lieder GmbH & Co. KG, Ludwigsburg/Germany) who attained consent from the subjects included.

**Does the dataset relate to people?**

Yes. The dataset consists of microscopic images of human white blood cells.

**Did you collect the data from the individuals in question directly, or obtain it via third parties or other sources (e.g., websites)?**

Data have not been obtained via third parties.

**Were the individuals in question notified about the data collection?**

As the blood smear slides were bought from a company, notification to individuals of the data collection has been performed by the company.

**Did the individuals in question consent to the collection and use of their data?**

Yes, they did.

**If consent was obtained, were the consenting individuals provided with a mechanism to revoke their consent in the future or for certain uses?**

We do not know the conditions of consent adopted by the selling company. However, we believe the company provided the individuals a complete freedom in revoking their consent in the future, if desired.

**Has an analysis of the potential impact of the dataset and its use on data subjects (e.g., a data protection impact analysis) been conducted?**

No, this kind of analysis has not been conducted.

---

**Preprocessing/cleaning/labeling**

---

**Was any preprocessing/cleaning/labeling of the data done (e.g., discretization or bucketing, tokenization, part-of-speech tagging, SIFT feature extraction, removal of instances, processing of missing values)?**

Intensity scaled images are generated with Jetraw Data Suite for both datasets, which applies a physical model based on sensor calibration to accurately simulate intensity reduction. Microscopy Raw images are extracted from RGB Microscopy data through a pixel selection from images taken with three filters, in order to have a Bayer Pattern. Pixels intensities are rescaled with Jetraw Data Suite to match the measured transmissivities of a Bayer colour filters array.

**Was the "raw" data saved in addition to the preprocessed/cleaned/labeled data (e.g., to support unanticipated future uses)?**

Raw images are available in the dataset.

**Is the software used to preprocess/clean/label the instances available?**

All code used in the experiments of this manuscript is publicly available. Jetraw products that were used for acquiring the data are commercially available.

---

**Uses**

---

**Has the dataset been used for any tasks already?**

The dataset has not yet been used.

**Is there a repository that links to any or all papers or systems that use the dataset?**

The repository at `https://github.com/aiaudit-org/raw2logit` associated to this work, maintained by Luis Oala.

**What (other) tasks could the dataset be used for?**

The dataset can be used to study the effect of image signal processing on the performance and robustness of any other machine learing model implemented in PyTorch, designed for a supervised multiclass classification task.

**Is there anything about the composition of the dataset or the way it was collected and pre-processed/cleaned/labeled that might impact future uses?**

To the best of our knowledge, we do not recognize such impacts.

**Are there tasks for which the dataset should not be used?**

To the best of our knowledge, there are no such tasks.

---

**Distribution**

**Will the dataset be distributed to third parties outside of the entity (e.g., company, institution, organization) on behalf of which the dataset was created?**

Yes. The dataset will be publicly available.

**How will the dataset will be distributed (e.g., tarball on website, API, GitHub)**

A guide to access the dataset is available at `https://github.com/aiaudit-org/raw2logit`. Moreover, the dataset can be downloaded anonymously and directly at `https://zenodo.org/record/5235536` under the doi: 10.5281/zenodo.5235536.

**When will the dataset be distributed?**

The dataset is already publicly available.

**Will the dataset be distributed under a copyright or other intellectual property (IP) license, and/or under applicable terms of use (ToU)?**

The dataset will be distributed under the Creative Commons Attribution 4.0 International.

**Have any third parties imposed IP-based or other restrictions on the data associated with the instances?**

No.

**Do any export controls or other regulatory restrictions apply to the dataset or to individual instances?**

There are no such restrictions.

---

**Maintenance**

**Who will be supporting/hosting/maintaining the dataset?**

Luis Oala on behalf of Dotphoton AG.

**How can the owner/curator/manager of the dataset be contacted (e.g., email address)?**

By email address via luis.oala@dotphoton.com.

**Is there an erratum?**

At the time of submission, there is no such erratum. If an erratum is needed in the future it will be accessible at `https://github.com/aiaudit-org/raw2logit`.

**Will the dataset be updated (e.g., to correct labeling errors, add new instances, delete instances)?**

Yes. The dataset will be enlarged wrt. the number of instances.

**If the dataset relates to people, are there applicable limits on the retention of the data associated with the instances (e.g., were individuals in question told that their data would be retained for a fixed period of time and then deleted)?**

To the best of our knowledge, there are no such limits.

**Will older versions of the dataset continue to be supported/hosted/maintained?**

Older versions will be supported and maintained in the future. The dataset will continue to be hosted as long as `https://zenodo.org/` exists.

**If others want to extend/augment/build on/-contribute to the dataset, is there a mechanism for them to do so?**

For any of these requests contact either Luis Oala (luis.oala@dotphoton) or Bruno Sanguinetti (bruno.sanguinetti@dotphoton.com). For now, we do not have an established mechanism to handle these requests.

| Composition of Raw-Microscopy | |
|---|---|
| Type of instances | Image and label |
| Objects on images | White blood cells |
| Type of classes | Morphological classes |
| Number of instances | 940 |
| Number of classes | 9 |
| Image size | 256 by 256 pixels |
| Image format | `.tif` |
| Raw image format | Please see Section 4.1 |

Table 12: A summary of the composition of Raw-Microscopy.

**Datasheet for Raw-Drone**

---

| Motivation |
|:----------:|

**For what purpose was the dataset created?**

With Raw-Drone we provide a publicly available raw dataset in order to examine the effect of the image data processing on the performance and the robustness of machine learning models. This dataset enables to study these effects for a segmentation task: the segmentation of cars. The dataset was taken with specified parameters: sensor gain, point-spread function and ground-sampling distance, so that physical models may be used to process the data. It also was taken with a easily accessible and affordable system, so that it may be reproduced.

**Who created this dataset (e.g., which team, research group) and on behalf of which entity (e.g., company, institution, organization)?**

The dataset was created by Bruno Sanguinetti and Marco Aversa on behalf of the company Dotphoton AG.

**Who funded the creation of the dataset?**

The data collection was funded by Dotphoton AG. The calibration of the image characteristics was jointly funded by Dotphoton AG and the European Space Agency.

| Composition |
|:-----------:|

**What do the instances that comprise the dataset represent (e.g., documents, photos, people, countries)?**

An instance is a tuple of an image and a segmentation mask. The image shows a landscape shot from above. The segmentation mask is a binary image. A white pixel in this mask corresponds to a pixel within a region in the image where a car is displayed. A black pixel in this mask corresponds to a pixel within a region in the image where no car is displayed.

**How many instances are there in total (of each type, if appropriate)?**

The dataset consists of 548 instances.

**Does the dataset contain all possible instances or is it a sample (not necessarily random) of instances from a larger set?**

The dataset does not contain all possible instances. Only images with at least one white pixel in the associated segmentation mask are considered.

**What data does each instance consist of? "Raw" data (e.g., unprocessed text or images) or features?**

Both, the image and the segmentation mask consist of 256 by 256 pixels. The image is a raw image in `.tif` format and the the segmentation mask is in `.png` format. The images are cropped sub-images of 12 raw images in `.DNG` format, consisting of 3648 by 5472 pixels.

**Is there a label or target associated with each instance?**

Each instance is associated to a binary segmentation mask.

**Is any information missing from individual instances?**

No information is missing.

**Are relationships between individual instances made explicit (e.g., users' movie ratings, social network links)?**

Since every image is a cropped sub-image of an original image, several of these sub-images belong to the same original image. All sub-images are disjoint, i.e. no different images share a pixel from the original image.

**Are there recommended data splits (e.g., training, development/validation, testing)?**

There are no recommended data splits. All the data splits that we used for our experiments were randomly picked.

**Are there any errors, sources of noise, or redundancies in the dataset?**

To the best of our knowledge, there are no errors in the dataset. The segmentation mask is created by hand and hence noisy, especially at the boundaries between a region with a car and a region without a car.

**Is the dataset self-contained, or does it link to or otherwise rely on external resources (e.g., websites, tweets, other datasets)?**

The dataset is self-contained.

**Does the dataset contain data that might be considered confidential (e.g., data that is protected by legal privilege or by doctor-patient confidentiality, data that includes the content of individuals non-public communications)?**

No. The dataset does not contain data of any of the above types.

**Does the data set contain data that, if viewed directly, might be offensive, insulting, threatening, or might otherwise cause anxiety?**

No. The dataset does not contain data with any of the above properties.

**Does the dataset relate to people?**

The dataset does not relate to people. The drone data was screened for PIIs such as faces or license plates on cars and removed by the data collection team.

**Any other comments?**

See table 13 for a summary of the composition of the Raw-Drone.

---



**Collection Process**



---

**How was the data associated with each instance acquired?**

The data was collected by flying a drone and saving the raw data. The calibration data for the drone's imager was acquired both under laboratory conditions and using a ground-based calibration target, so that it could be acquired under operating conditions.

**What mechanisms or procedures were used to collect the data (e.g., hardware apparatus or sensor, manual human curation, software program, software API)?**

To acquire the drone images, we used a DJI Mavic 2 Pro Drone, equipped with a Hasselblad L1D-20c camera (Sony IMX183 sensor). This system has $2.4\,\mu m$ pixels in Bayer filter array. Images were taken with the drone hovering for maximum stability. This stability was verified to be better than a single pixel by calculating the correlation of subsequent images. The

objective has a focal length of $10.3\,mm$. We operated this objective at an f-number of $N = 8$, to emulate the PSF circle diameter relative to the pixel pitch and ground sampling distance (GSD) as would be found on images from high-resolution satellites. Operating at $N = 8$ also minimises vignetting, aberrations, and increases depth of focus. The point-spread function (PSF) was measured to have a circle diameter of $12.5\,\mu m$ using the edge-spread function technique and a ground calibration target. This corresponds to $\sigma = 2.52\,px$, which also corresponds to a diffraction-limited system, within the uncertainty dictated by the wavelength spread of the image. Images were taken at $200\,ISO$, corresponding to a gain of $0.528\,DN/e^-$. The 12-bit pixel values are however left-justified to 16-bits, so that the gain on the 16-bit numbers is $8.448\,DN/e^-$. The images were taken at a height of $250\,m$, so that the GSD is $6\,cm$. All images were tiled in 256x256 patches. Segmentation color masks were created to identify cars for each patch. From this mask, classification labels were generated to detect if there is a car in the image. The dataset is constituted by 548 images for the segmentation task, and 930 for classification. Six additional intensity scales were created with Jetraw.

**If the dataset is a sample from a larger set, what was the sampling strategy (e.g., deterministic, probabilistic with specific sampling probabilities)?**

The entire dataset is presented.

**Who was involved in the data collection process (e.g., students, crowdworkers, contractors) and how were they compensated (e.g., how much were crowdworkers paid)?**

The dataset was taken by a company employee, compensated by his salary. Labeling was performed by both a company employee and a PhD student, who's PhD is funded by the company.

**Over what timeframe was the data collected? Does this timeframe match the creation timeframe of the data associated with the instances (e.g., recent crawl of old news articles)?**

The dataset was taken as the initial step of writing this article.

**Were any ethical review processes conducted (e.g., by an institutional review board)?**

The dataset does not contain any elements requiring an ethical review process.

**Does the dataset relate to people?**

The dataset does not relate to people. There are individuals on the images, but it is not possible to identify these individuals.

**Preprocessing/cleaning/labeling**

**Was any preprocessing/cleaning/labeling of the data done (e.g., discretization or bucketing, tokenization, part-of-speech tagging, SIFT feature extraction, removal of instances, processing of missing values)?**

No further processing was applied to the Raw-Drone data.

**Was the "raw" data saved in addition to the preprocessed/cleaned/labeled data (e.g., to support unanticipated future uses)?**

Raw images are available in the dataset.

**Is the software used to preprocess/clean/label the instances available?**

All code used in the experiments of this manuscript is publicly available. Jetraw products that were used for acquiring the data are commercially available.

**Uses**

**Has the dataset been used for any tasks already?** The dataset has not yet been used.

**Is there a repository that links to any or all papers or systems that use the dataset?**

The repository at `https://github.com/aiaudit-org/raw2logit` associated to this work, maintained by Luis Oala.

**What (other) tasks could the dataset be used for?**

The dataset can be used to study the effect of image signal processing on the performance and robustness of any other machine learing model implemented in PyTorch, designed segmentation task.

**Is there anything about the composition of the dataset or the way it was collected and preprocessed/cleaned/labeled that might impact future uses?**

To the best of our knowledge, we do not recognize such impacts.

**Are there tasks for which the dataset should not be used?**

To the best of our knowledge, there are no such tasks.

**Distribution**

**Will the dataset be distributed to third parties outside of the entity (e.g., company, institution, organization) on behalf of which the dataset was created?**

Yes. The dataset will be publicly available.

**How will the dataset will be distributed (e.g., tarball on website, API, GitHub)**

A guide to access the dataset is available at `https://github.com/aiaudit-org/raw2logit`. Moreover, the dataset can be downloaded anonymously and directly at `https://zenodo.org/record/5235536` under the doi: 10.5281/zenodo.5235536.

**When will the dataset be distributed?**

The dataset is already publicly available.

**Will the dataset be distributed under a copyright or other intellectual property (IP) license, and/or under applicable terms of use (ToU)?**

The dataset will be distributed under the Creative Commons Attribution 4.0 International.

**Have any third parties imposed IP-based or other restrictions on the data associated with the instances?**

No.

**Do any export controls or other regulatory restrictions apply to the dataset or to individual instances?**

There are no such restrictions.

**Maintenance**

**Who will be supporting/hosting/maintaining the dataset?**

Luis Oala on behalf of Dotphoton AG.

**How can the owner/curator/manager of the dataset be contacted (e.g., email address)?**

By email address via luis.oala@dotphoton.com.

**Is there an erratum?**

At the time of submisson, there is no such erratum. If an erratum is needed in the future it will be accessible at `https://github.com/aiaudit-org/raw2logit`.

**Will the dataset be updated (e.g., to correct labeling errors, add new instances, delete instances)?**

Yes. The dataset will be enlarged wrt. the number of instances.

**If the dataset relates to people, are there applicable limits on the retention of the data associated with the instances (e.g., were individuals in question told that their data would be retained for a fixed period of time and then deleted)?**

To the best of our knowledge, there are no such limits.

**Will older versions of the dataset continue to be supported/hosted/maintained?**

Older versions will be supported and maintained in the future. The dataset will continue to be hosted as long as `https://zenodo.org/` exists.

**If others want to extend/augment/build on/contribute to the dataset, is there a mechanism for them to do so?**

For any of these requests contact either Luis Oala (luis.oala@dotphoton.com) or Bruno Sanguinetti (bruno.sanguinetti@dotphoton.com). For now, we do not have an established mechanism to handle these requests.

| Composition of Raw-Drone | |
|---|---|
| Type of instances | Image and mask |
| Objects on images | Landscape shots from above |
| Number of instances | 548 |
| Number of original images | 12 |
| Image size | 256 by 256 pixels |
| Mask size | 256 by 256 pixels |
| Original image size | 3648 by 5472 |
| Image format | `.tif` |
| Mask format | `.png` |
| Raw image format | `.DNG` |

Table 13: A summary of the composition of Raw-Drone.