# OpenReview forum: "Data Models for Dataset Drift Controls in Machine Learning With Optical Images"
_TMLR — Accepted by TMLR_

### Review · Reviewer_XUNK · 2023-02-22

**Summary Of Contributions:**

- The authors propose a new framework for dataset drift for multiple use-cases: robustness evaluation, data model optimization, etc.
- The authors show some interesting experimental results on both robustness evaluation and data model optimization. Especially, the data model optimization is interesting that it can further improve the downstream performance significantly.

**Audience:**

Yes

**Claims And Evidence:**

Yes

**Requested Changes:**

1. Figure 1
- It would be great if the authors can explain more about drift synthesis, drift forensics and drift optimization.
- The explanation of those three things (in page 4) is far from the Figure. Good to add some brief summaries.

2. Figure 4 and Figure 5
- What is the meaning of comparing ISP and common corruption benchmarks?
- The conclusion for each left and right figure is different. Left figure shows the robustness of the physically faithful one and the right figure shows the robustness of the physically non-faithful one.
- When we test the robustness, we may check both parts. For instance, brightness can commonly occur in dataset shifts and it would be also important to make a robust model on the brightness.
- Also, blur is another important part in the Autonomous vehicle case (imagine the rainy day).
- So, it is unclear why the authors compare those two as side by side even though those are not apple to apple comparisons.

3. Limitations & Assumption
- As we know, the authors assume that we have the raw sensor data. However, this is not a common setting and it is very hard to find the public datasets with raw sensor data.
- It would be good if the authors state this assumption in the earlier part of the paper (much earlier than page 6).

4. Paper layout
- Personally, some parts of the paper can be moved to the appendix to make this paper more concise.
- For instance, the details of Section 4.1 can be moved to the appendix.
- Majority of Section 4.2.1 and 4.2.2 can be moved to the appendix.

5. Etc
- Figure 2: Microscopy should be columns one to two.
- It would be great if the authors use the consistent term in the paper. For instance, in some cases, the authors use "regression / segmentation" interchangeably.

**Strengths And Weaknesses:**

Strength
- The proposed framework is novel and under-explored.
- Data model optimization part is well supported by the experiments.

Weakness
- It is unclear why we should compare physically faithful one and physically non-faithful one as side by side in Figure 4 and 5.
- The paper length can be significantly reduced (and make it more concise).
- Limitation and assumptions should be clearly clarified and should be mentioned much earlier.

---

> ### Author Response · Authors · 2023-02-26
> **Paper updated with your suggestions**
>
> Dear reviewer XUNK,
>
> Thank you for your suggestions! We incorporated them and uploaded the updated manuscript (https://openreview.net/pdf?id=I4IkGmgFJz). Note that we made slight changes to text in blue so you can find them easily.
>
> > *Figure 1
> > It would be great if the authors can explain more about drift synthesis, drift forensics and drift optimization.
> > The explanation of those three things (in page 4) is far from the Figure. Good to add some brief summaries.*
> * added short descriptions directly in the figure captions for easier reading
>
> > *Figure 4 and Figure 5
> > What is the meaning of comparing ISP and common corruption benchmarks?
> > The conclusion for each left and right figure is different. Left figure shows the robustness of the physically faithful one and the right figure shows the robustness of the physically non-faithful one.
> > When we test the robustness, we may check both parts. For instance, brightness can commonly occur in dataset shifts and it would be also important to make a robust model on the brightness.
> > Also, blur is another important part in the Autonomous vehicle case (imagine the rainy day).
> > So, it is unclear why the authors compare those two as side by side even though those are not apple to apple comparisons.*
> We added a new paragraph to section 5.1.1 to explain why the comparison is made.
>
> We agree with you that a direct apple-to-apple comparison is not possible. This is part of the point we are making: post-hoc augmentations are not physically faithful, hence they cannot be related to real-world variations in data models and consequently lead to wrong conclusions for model selection.
>
> As you point out, in machine learning practice, augmentation users often assume that applying a corruption, for example 'blur', to a processed image will emulate the noise from a real-world camera system, for example blur from the lens or the denoising component in the camera. However, this is not the case. Similarly,  variations in the physically accurate sources of brightness of an image (such as exposure time, black level or white balancing) are not equivalent to changing the brightness of a processed image post-hoc. This can be gleaned from the composition of the optical data generating process (see Figure 1 and Sections 4.2.1 and 4.2.2) and has also been empirically demonstrated in previous work (see for example references 114-116 in our manuscript). Here we go one step further to show that physically *un*faithful augmentation testing can lead to wrong conclusions in model selection.
>
> We are only referring to limitations of post-hoc augmentations relating to robustness testing and model selection. Augmentations have important empirically validated benefits in other applications such as regularization during training or semi- and self-supervised learning. This we also mention in the manuscript.
>
> Do let us know if something remains unclear!
>
> > *Limitations & Assumption
> > As we know, the authors assume that we have the raw sensor data. However, this is not a common setting and it is very hard to find the public datasets with raw sensor data.
> > It would be good if the authors state this assumption in the earlier part of the paper (much earlier than page 6).*
> * Sure, we added this constraint explicitly to the abstract as well as an additional paragraph to section 4.
>
> > *Paper layout
> > Personally, some parts of the paper can be moved to the appendix to make this paper more concise.
> > For instance, the details of Section 4.1 can be moved to the appendix.
> > Majority of Section 4.2.1 and 4.2.2 can be moved to the appendix.*
> * Good suggestion, we moved the details of sections 4.1, 4.2.1 and 4.2.2 to the appendices.
>
> > *Figure 2: Microscopy should be columns one to two.*
> * We count the label columns (i.e. \$ \boldsymbol{y}\$) as individual columns (both for microscopy and drone). If this is confusing let us know and we can add some numbers.
>
> > *It would be great if the authors use the consistent term in the paper. For instance, in some cases, the authors use "regression / segmentation" interchangeably.*
> * Yes, we treat the segmentation task with binary cross-entropy loss as a concrete instance of regression. If there are instances were this is confusing we can adjust it.

---

### Review · Reviewer_R3iq · 2023-03-04

**Summary Of Contributions:**

This is a revision of a previously reviewed submission. Briefly, it proposes modelling the image processing pipeline in imaging devices to help with simulating and recovering from domain shifts due to changes in the image processing.


**Audience:**

Yes

**Claims And Evidence:**

No

**Requested Changes:**

- Clarify the use-cases and motivation in the setup where RAW data is available
- Provide more comprehensive experiments on RAW data for section 5.3 - multiple runs and model tuning for RAW data
- Include a discussion and if possible, comparison using reconstructed RAW data
- [Optional] Analyze the effect of incorrect model selection on actual task performance for section 5.1


**Strengths And Weaknesses:**

The requested changes by the AE were partially addressed and the paper is improved. However, some issues remain unaddressed. Overall it would be helpful if all changes over the last revision were highlighted for ease of reference.

Suggestion 1:
- I may have missed something, but it did not appear to me there were any changes to the introduction (highlighted in blue), beyond the caption of Figure 1.
- It is good that a separate section discussing potential use cases and limitations is included for each of the three applications. However, it still remains unclear what the value of modelling the image processing pipeline is for drift forensics if RAW data is avaliable - any shifts due to image processing can be avoided by training on the RAW data; modelling the image processing pipeline will not help for shifts due to the sensor or optics, which are not captured in the pipeline. Similarly, the motivation for drift synthesis is again not clear if RAW data is available at both sites. The exact setup and use-case should be further clarified.

Suggestion 2:
- It does not appear that additional experiments were included beyond those included in the revision for the original submission as Appendix A.4.3; this result has been moved into the main text, which is good. However, more extensive evaluation should be done on the RAW data - for instance, at least 3 cross-validation runs per dataset, and task models should be tuned for the RAW data; details on the RAW data training should be included in A.2. The caption for Figure 7 should also be updated accordingly.
- I did not see an expanded discussion on RAW training in Section 6.

Suggestion 3:
- The changes to Section 5.1 address my concerns. One finer point which would improve the paper would be to consider the actual impact of incorrect model selection on performance, by comparing the difference in performance on the shifted data when using the best model selecting using physically-faithful and post-hoc corruption cases.
- On inverting processing - the point is that if the reconstructed RAW data is sufficiently good for training models that generalize, then modelling the image processing pipeline may not be necessary. As such it is a relevant comparison and should be at least discussed in the paper.

Suggestion 4:
- While the processing may be done in silico, there will likely be sensor noise which has to be accounted for in any real application. Using real data will give a sense of this.

---

> ### Author Response · Authors · 2023-03-07
> **Update**
>
> Dear Reviewer R3iq,
>
> Thanks for your effort and feedback!
>
> We are incorporating your suggestions and will update here soon.
>
> Re the color code: ah that is unfortunate. What you saw in blue is an already updated version to this review round in response to reviewer XUNK. It seems openreview kept this interaction hidden from you until now but you were able to download the updated manuscript. We are incorporating your suggestions with ergonomic color codes so you should be able to conveniently trace changes.
>
> In good spirits,
> The authors

---

> ### Author Response · Authors · 2023-03-25
> **Documentation of changes made**
>
> Dear reviewer R3iq,
>
> We updated the paper to incoporate your suggestions. Thanks again for the great feedback! We hope the updates address your questions.
>
> # Requested changes
>
> Here is a digest of the changes you asked for. Detailed explanations of these changes are in our detailed responses below.
>
> > *Clarify the use-cases and motivation in the setup where RAW data is available*
>
> Added section 1.3 in the introduction on "Scope, practical fit and limitations".
>
> > *Provide more comprehensive experiments on RAW data for section 5.3 - multiple runs and model tuning for RAW data*
>
> Added the results and documentation you suggested.
>
>
> > *Include a discussion and if possible, comparison using reconstructed RAW data*
>
> A paragraph was added in section 4.1 to explain why the study does not use reconstructions, citing the inability to guarantee physical faithfulness and providing pointers to related works for application-specific performance boosts.
>
> > *[Optional] Analyze the effect of incorrect model selection on actual task performance for section 5.1*
>
> These results can be gleaned from the detailed ranking tables in A.3.1. We added more detail in our reply above.
>
> # Detailed responses
>
> ## Suggestion 1
>
> > *I may have missed something, but it did not appear to me there were any changes to the introduction (highlighted in blue), beyond the caption of Figure 1.*
>
> Ah, that is unfortunate. What you saw in blue is an already updated version to this review round in response to reviewer XUNK. It seems openreview kept this interaction hidden from you until now but you were nevertheless able to download the updated manuscript. The new results were also uploaded to the original submission (ranking results, raw results, model selection results). New consolidated color codes for better navigation are:
>
> * Green -  changes from the original submission
> * Blue - reviewer XUNK changes
> * Purple - reviewer R3iq (you) changes
> * Red - reviewer Fxgr changes
>
> > *It is good that a separate section discussing potential use cases and limitations is included for each of the three applications [...]*
>
> We added a section 1.3 in the introduction where we address your questions and summarize the scope, practical fit and limitations of the proposed method in one place.
>
> ## Suggestion 2:
>
> > *It does not appear that additional experiments were included beyond those included in the revision for the original submission as Appendix A.4.3; this result has been moved into the main text, which is good. [...]*
>
> We added requested results (3 fold CV) and updated Figure 7 caption. Task models are always trained in the experiments. We made an additional note of it in section 5.3.3 + reference to A.2 for the raw training.
>
> > *I did not see an expanded discussion on RAW training in Section 6.*
> We added section 5.3.3 ("Raw and data models") and expanded the last two paragraphs of the discussion related to RAW data. We hope it provides a useful contextualization.
>
> ## Suggestion 3:
>
> > *The changes to Section 5.1 address my concerns. One finer point which would improve the paper would be to consider the actual impact of incorrect model selection on performance, by comparing the difference in performance on the shifted data when using the best model selecting using physically-faithful and post-hoc corruption cases.*
>
> These results can be gleaned from the detailed ranking tables in A.3.1. For example, the best performing microscopy data-task-model combo selected across all test ISPs [(ma,s,me) with .83 avg accuracy] has more than 20 percentage points gap compared Common Corruptions (.62 avg accuracy) (see Table 4 for more comparisons).
>
>
> > *On inverting processing - the point is that if the reconstructed RAW data is sufficiently good for training models that generalize, then modelling the image processing pipeline may not be necessary. As such it is a relevant comparison and should be at least discussed in the paper.*
>
> We added a paragraph in section 4.1 to explain why we are not using reconstructions. Our goal is physical faithfulness of the data which cannot be guaranteed by reconstructions, as existing work that we cite demonstrates. Going beyond physically faithful drift controls, for example application specific performance boosts,  the cited works provide good pointers to related work.
>
> ## Suggestion 4:
>
> > *While the processing may be done in silico, there will likely be sensor noise which has to be accounted for in any real application. Using real data will give a sense of this.*
>
> We agree with you that sensor noise as well as other components related to hardware, for example the lens, are relevant additional factors for data drift. However, we are clear and upfront about the scope of this work as limited to the ISP.
>
> In good spirits,
> The authors

---

> > ### Comment · Reviewer_R3iq · 2023-04-17
> > **Thanks for the revisions**
> >
> > Thanks for the revisions. Here are some further comments.
> >
> > **Suggestion 1**
> >
> > The newly included Section 1.3 did help me understand where this paper is coming from, but still left me with the following question: How do sensor and optics drifts compare to the modeled drifts? Claims about the practical utility of the current version of the pipeline depend on this.
> >
> > **Suggestion 2**
> >
> > Numerical scores for the comparison between various models and tasks should be included as a table in the main text as it is not really possible to see the differences from the plots especially for the comparison with training on RAW data.
> >
> > Regarding the discussion, it seems to me the end-game should be training on RAW and that the proposed pipeline is an interim solution while RAW data becomes more widely available. Perhaps this should be included in the discussion as well, but I leave it up to the authors to decide.
> >
> > **Suggestion 3**
> >
> > > These results can be gleaned from the detailed ranking tables in A.3.1. For example, the best performing microscopy data-task-model combo selected across all test ISPs [(ma,s,me) with .83 avg accuracy] has more than 20 percentage points gap compared Common Corruptions (.62 avg accuracy) (see Table 4 for more comparisons).
> >
> > Yes the results are in the paper, but again explicitly making the comparison as in the response can help to explain the benefit/reason for doing a comparison between these disparate cases (e.g. Reviewer XUNK's comments).
> >
> > > We added a paragraph in section 4.1 to explain why we are not using reconstructions. Our goal is physical faithfulness of the data which cannot be guaranteed by reconstructions, as existing work that we cite demonstrates. Going beyond physically faithful drift controls, for example application specific performance boosts, the cited works provide good pointers to related work.
> >
> > It may not be physically faithful but the outputs from such methods nonetheless can be used to perform the tasks described in the paper. It is of interest to readers to know how the proposed methods compare to put the work into context, especially as they require special acquisition of RAW images to calibrate the data model in the first place (see also the comment from Reviewer Fgxr on gamma correction).
> >
> > **Suggestion 4**
> >
> > This relates to the practical utility of the current method, and a previous question under (Suggestion 1) - how much is the sensor/optical drift compared to ISP drift in real-use cases? If some discussion on this point could be included I think my concern would be addressed.

---

> > > ### Author Response · Authors · 2023-04-17
> > > **Thank you + updates**
> > >
> > > Dear reviewer R3iq,
> > >
> > > Thank you for the response, we will update the final points in line with your suggestions in the next two days.
> > >
> > > In good spirits,
> > > The authors
> > >
> > > --EDIT April 25, 2023--
> > >
> > > Thank you again for your great comments! We uploaded an updated manuscript.
> > >
> > > > **Suggestion 1**
> > >
> > > Thank you, we added further discussion to section 1.3, also considering your comment regarding *Suggestion 4*
> > >
> > > > **Suggestion 2**
> > >
> > > A table following your suggestions was added for clarity.
> > >
> > > Good framing, we added your "endgame/interim" suggestion to the discussion insection 5.3.4.
> > >
> > > > **Suggestion 3**
> > >
> > > Got it, very good point to make it as explicit as possible, we added it to text + same comparison for segmentation where it is data model (ma,u,ga).
> > >
> > > Since ISPs introduce loss of information as existing work [178, 179] has extensively studied, it is not clear how one can expect to recover the original raw image. Introduction of this uncertainty, we agree a highly interesting subject in its own right and in particular as relating to inverse problems, is orthogonal to the goal of precise physical faithfulness of the methods presented here. However, following your suggestion, for interested readers we added further contextualization at the end of section 4.1 what performance gains can be expected under a physically unfaithful regime.
> > >
> > > > **Suggestion 4**
> > >
> > > See our comment on *Suggestion 1*

---

### Review · Reviewer_Fgxr · 2023-03-06

**Summary Of Contributions:**

The paper proposes the use of explicit data models for the processing of raw (optical) imaging data to 1) enable insights into the performance of a "task model" under "more realistic" data drift; 2) the analysis of vulnerabilities of the task model in  terms of data processing; 3) the joint optimisation of the data processing pipeline together with the task model.

I believe the usage of explicit modelling of the data processing pipeline to be novel and interesting to the community.

**Audience:**

Yes

**Broader Impact Concerns:**

Nothing major.

**Claims And Evidence:**

Yes

**Requested Changes:**

Please expand on the way the data models would be used in practice and what requirements go hand-in-hand with these scenarios.

**Strengths And Weaknesses:**

Pros:
- The paper introduces a novel and interesting way to think about the preprocessing in computer vision problems that allows for a more realistic (but limited) characterisation of data drift.
- The authors collect and release novel datasets allowing the study of data preprocessing pipelines on raw images.
- The paper provides an interesting comparison of the data drifts induced by the proposed data models as well as more commonly used data augmentations.

Cons:

- In the current form of the paper, it is not clear in which scenarios one would use a data model as proposed. The authors show three different use cases in terms of 1) "drift analysis", 2) "drift forensics" and 3) "drift optimisation". However, it is not clear how useful those applications are in practice.
  - 1) assumes that one would build a data model to predict the difference between sites `s` and `t` from only using data acquired on site `s`. However, this assumes that the data model for `t` is fixed and one does not have access to the raw data. Furthermore, it is only really useful for characterising the changes due to the data model and does not capture changes due to the optics or other drifts.
  - 2) assumes that it is interesting to understand a models vulnerabilities due to changes in  the data preprocessing. This is only relevant if the model does not "contain" the data processing pipeline as well. But the paper seems to argue that raw data should always be considered when modelling a task.
  - 3) This is potentially the most useful one. However, it is possible to also optimise the processing of "preprocessed data" e.g. the parameters of the gamma correction. Furthermore, it would then lack comparisons to works optimising the preprocessing such as [1].
- The data model is currently very rigid in it's form. It would be interesting to expand upon this by considering different orders of processing steps. It also is not clear where this exact pipeline comes from and how it relates to common image processing pipelines implemented by, e.g. camera manufacturers.

Minor:
- I'd suggest changing the title to make it clear this work is limited to "optical images".
- Introduction: It is said the gradient of the data model allows for "precise tolerancing" of the task model sensitivity. What's meant by that?
- What's eq. 1 supposed to convey? I'm not sure how to read the equation.
- Sec 1.1. explains the need for physically realistic data models. However, I'd argue that public APIs might be exposed to changes that are not physically reasonable. Furthermore, it lacks any mention of potential changes to $x_{raw}$ due to changes in optics or image acquisition or otherwise. E.g. the drone dataset could be exposed to data drift because of seasonal changes to the environment.
- Sec 1.1. - it is argued that adding Gaussian noise is not physically realistic. However, could this be used for modelling sensor noise either by adding it to $x_{raw}$ or the processed image.
- The contribution mention that the drone dataset is a regression dataset - however it seems to be a segmentation one?
- How are data models etc related to data normalisation?
- Sec 4.1. mentions that the datasets are collected keeping "positive impact" in mind. However, I'd be careful with the drone dataset.
- Sec 4.1. assumes that images are always RGB images. How could this be extended to greyscale images or images where multiple "channels" are captured with specific colour filters? (This isn't uncommon in biology or potentially astronomy or others)
- Sec 4.1. - I believe it might be clearer to say "The lens has a focal length of ..." instead of "objective".
- Sec. 4.1. - What are segmentation color masks?
- Fig 3. could use a general caption.
- Sec 4.2.1 "in 3a" should be "in Fig. 3a"
- Why is color correction performed in YUV space? Is the color conversion differentiable?
- Why do you define the parameter space sets for every operation?
- Fig 4: How come the diagonal isn't best for a given data model?
- Fig 4: There seems to be a mismatch with the "*". ma,u,me is the best on "identity" on the right, but not the best for "ma,u,me" test pipeline.
- Fig 4: How is the worst case defined?
- Fig 4: How do you take the average when saying "the average performance ... drops ..."?

- Fig 6.c: What's the forensics loss?
- Sec 5.2.3: Why would the data model not be part of the "shipped model"?
- Fig 7: What does "Raw comparison" refer to? Is this the model trained on raw data? If so, it should  be integrated into the the other plots so that it is easy to compare. How is the raw model trained? For segmentation, does the target segmentation mask need to be processed to fit the raw data space before demosaicing?
- Fig 7: It is said this isn't possible to run similar optimisation on processed data. However, it would be possible to tune hyperparameters of gamma correction or similar. Furthermore, there is work on learning data normalisation in medical imaging [1]

[1] Drozdzal, Michal, et al. "Learning normalized inputs for iterative estimation in medical image segmentation." Medical image analysis 44 (2018): 1-13.

---

> ### Author Response · Authors · 2023-03-07
> **Update**
>
> Dear Reviewer Fgxr,
>
> Thanks for your effort and feedback!
>
> We are incorporating your suggestions and will update here soon.
>
> In good spirits,
> The authors
>
> Edit March 25, 2023: First revisions have been uploaded to openreview. We will post detailed response in the coming days. Thank you for your patience. Due to ICML rebuttal period overlap it took us a bit longer to incorporate all changes.

---

> > ### Author Response · Authors · 2023-03-27
> > **Responses and documentation of changes made (1/3)**
> >
> > Dear reviewer Fgxr,
> >
> > We would like to express our gratitude for your considerate and detailed review. Please find our responses below. An updated manuscript has also been uploaded to openreview. For easier navigation the following color codes are used:
> >
> > * Green - changes from the original submission
> > * Blue - reviewer XUNK changes
> > * Purple - reviewer R3iq changes
> > * Red - reviewer Fxgr (you) changes
> >
> > # Requested Changes
> > > *Please expand on the way the data models would be used in practice and what requirements go hand-in-hand with these scenarios.*
> >
> > We added a section 1.3 in the introduction where we address your questions and summarize the scope, practical fit and limitations of the proposed method in one place.
> >
> > # Other questions
> > > *In the current form of the paper, it is not clear in which scenarios one would use a data model as proposed. The authors show three different use cases in terms of 1) "drift analysis", 2) "drift forensics" and 3) "drift optimisation". However, it is not clear how useful those applications are in practice.*
> >
> > Please see our clarification above.
> >
> > > *assumes that one would build a data model to predict the difference between sites s and t from only using data acquired on site s. However, this assumes that the data model for t is fixed and one does not have access to the raw data. Furthermore, it is only really useful for characterising the changes due to the data model and does not capture changes due to the optics or other drifts.*
> >
> > That is correct, the data models can only capture ISP related data drift which we explicitly state throughout the entire manuscript. We added section 1.3 to consolidate considerations on the scope and practicality of the data models.
> >
> > > *assumes that it is interesting to understand a models vulnerabilities due to changes in the data preprocessing. This is only relevant if the model does not "contain" the data processing pipeline as well. But the paper seems to argue that raw data should always be considered when modelling a task.*
> >
> > We assume the currently dominant setting in most application domains: access to raw data is possible but processed data is mainly produced and then fed into machine learning models. In this setting we propose different protocols to control drift due to processed data by exploiting raw interfaces in acquisition devices with potential to save cost and enable applications previously not possible.
> >
> > > *This is potentially the most useful one. However, it is possible to also optimise the processing of "preprocessed data" e.g. the parameters of the gamma correction. Furthermore, it would then lack comparisons to works optimising the preprocessing such as [1].*
> >
> > Fully agree and it possible (and done in the experiments) with the data models we present. For a sensitivity analysis of the task models wrt gamma correction only see Figure 6. Our framework however allows to flexibly capture the full ISP module on acquisition hardware which is not possible with processed data alone.
> >
> > Michal Drozdzal and colleagues' paper is great, however they do not address the physics of acquiring EM, CT, and MRI images nor do they work with raw sensor data or conduct physically accurate tests for drift validation, which is the main focus of our work.
> >
> > > *The data model is currently very rigid in it's form. It would be interesting to expand upon this by considering different orders of processing steps. It also is not clear where this exact pipeline comes from and how it relates to common image processing pipelines implemented by, e.g. camera manufacturers.*
> >
> > As there is a near combinatorial number of possible ISPs we picked common transformations and compositions to demonstrate the drift controls. "Common" is grounded in the textbook *Andy Rowlands. Physics of digital photography. IOP Publishing, 2017.* which we cite throughout ([122]).
> >
> > By design of the code base, individual functions in the data model (see */processing/pipeline_torch.py* and */processing/pipeline_numpy.py* in our code) can be swapped out, added or rearranged, enabling users to adjust the data model to their specific use cases.
> >
> > > *I'd suggest changing the title to make it clear this work is limited to "optical images".*
> >
> > Agreed it is more precise, we added "optical".

---

> > > ### Author Response · Authors · 2023-03-27
> > > **Responses and documentation of changes made (2/3)**
> > >
> > > > *Introduction: It is said the gradient of the data model allows for "precise tolerancing" of the task model sensitivity. What's meant by that?*
> > >
> > > Tolerancing is a term in engineering that describes a procedure by which the range in which a feature may vary is determined. Drift forensics utilize the upstream gradient to perform such an analysis on the data model.
> > >
> > > > *What's eq. 1 supposed to convey? I'm not sure how to read the equation.*
> > >
> > > We define drift as a change in distributions from $\mathcal{D}_t$ to $\mathcal{D}_s$ which is caused by a change in the random vector $\mathbf{V}$ as a result of a new data model $\tilde{\boldsymbol{\Phi}}$.
> > >
> > > > *Sec 1.1. explains the need for physically realistic data models. However, I'd argue that public APIs might be exposed to changes that are not physically reasonable. Furthermore, it lacks any mention of potential changes to due to changes in optics or image acquisition or otherwise. E.g. the drone dataset could be exposed to data drift because of seasonal changes to the environment.*
> > >
> > > We agree there are sources for drift outside the data models we provide and we are explicit about the scope and limits of the proposed framework. For example
> > > * Section 3: "*Here, common and core transformations are considered. Note that depending on the application context it is possible to reorder or add additional steps.*"
> > > * Section 6: "*. Furthermore, it should not be overlooked that dataset drift can also be caused by factors outside the ISP data model, for example the optical components of a camera. Our current data models are not yet capable of capturing factors that go beyond the ISP. Integrating work from lens manufacturing [133] to expand the reach explicit data models offers a promising next step for drift synthesis.*"
> > >
> > > To consolidate these assumptions and constraints early on in the manuscript we added section 1.3.
> > >
> > > > *Sec 1.1. - it is argued that adding Gaussian noise is not physically realistic. However, could this be used for modelling sensor noise either by adding it to or the processed image.*
> > >
> > > We argue that, as demonstrated in previous work, adding noise to a processed image is generally not equivalent to the noise model obtained by adding noise to an intermediate step of the data model which is then propagated through succeeding transformations of the data model. For the purposes of drift control this is relevant because existing corruption approaches provide no guarantees that the added noise is consistent with any physically plausible data model (see our formalization in Section 1.1). We do not argue against Gaussian noise per se but use it as example for the above distinction.
> > >
> > > > *The contribution mention that the drone dataset is a regression dataset - however it seems to be a segmentation one?*
> > >
> > > We use the binary cross entropy and Dice loss which are common loss of choice for image-to-image regression problems. More specifically for the case of binary segmentation masks the problem can be described as a pixel-wise logistic regression. That is why we use the terms segmentation and regression interchangeably.
> > >
> > > > *How are data models etc related to data normalisation?*
> > >
> > > You could view them as a form of normalisation to some extent: ensuring that raw data is processed through the same data model at train and deployment time prevents ISP related drift.
> > >
> > > > *Sec 4.1. mentions that the datasets are collected keeping "positive impact" in mind. However, I'd be careful with the drone dataset.*
> > >
> > > That is true. We added the Negative Societal Impact statement at the end to make clear that we do not allow our work be used for malicious purposes.
> > >
> > > > *Sec 4.1. assumes that images are always RGB images. How could this be extended to greyscale images or images where multiple "channels" are captured with specific colour filters? (This isn't uncommon in biology or potentially astronomy or others)*
> > >
> > > As long as the measurements can be directly mapped to a Bayer pattern the existing code can be used without modification. Otherwise, the primary operations in the code have to be adjusted.
> > >
> > >
> > > > *Sec 4.1. - I believe it might be clearer to say "The lens has a focal length of ..." instead of "objective".*
> > >
> > > Agreed, changed

---

> > > > ### Author Response · Authors · 2023-03-27
> > > > **Responses and documentation of changes made (3/3)**
> > > >
> > > > > *Sec. 4.1. - What are segmentation color masks?*
> > > >
> > > > The segmedntation masks for the cars are meant, we removed "color" to avoid confusion.
> > > >
> > > > > *Fig 3. could use a general caption.*
> > > >
> > > > Added caption
> > > >
> > > > > *Sec 4.2.1 "in 3a" should be "in Fig. 3a"*
> > > >
> > > > Thanks good spot! We missed a \Cref, it is fixed now.
> > > >
> > > > > *Why is color correction performed in YUV space? Is the color conversion differentiable?*
> > > >
> > > > YUV space is used as it is a common color model used in color imaging pipelines (see also our response to your question how the data models are motivated). Yes, all transformations are differentiable. See *self.colour_correction = nn.Parameter(torch.as_tensor(colour_matrix).reshape(3, 3))* in */processing/pipeline_torch.py* of our code which also has the default parameters on top.
> > > >
> > > > > *Why do you define the parameter space sets for every operation?*
> > > >
> > > > To make the notation precise and traceable to the code. We moved some of the technical notation to the appendices to improve the readability.
> > > >
> > > > > *Fig 4: How come the diagonal isn't best for a given data model?*
> > > >
> > > > Some data models appear to be superior data processors for a given task, for example (ma,u,ga), tying with or surpassing some training data models.
> > > >
> > > > > *Fig 4: There seems to be a mismatch with the "*". ma,u,me is the best on "identity" on the right, but not the best for "ma,u,me" test pipeline.*
> > > >
> > > > Good spot, but if you look closely there is a tie for the ISP test pipeline. Ties we resolved by comparing succeeding decimals. Full ranking results are in A.3. In fact, (ma,u,me) has the same scores for test pipeline and identity (.84) as to be expected.
> > > >
> > > > > *Fig 4: How is the worst case defined?*
> > > >
> > > > The globally worst scoring test environment for each setting (ISP and CC) were picked for qualitative illustration. Added the explanation to the figure caption.
> > > >
> > > > > *Fig 4: How do you take the average when saying "the average performance ... drops ..."?*
> > > >
> > > > We write "*The average performance of the task models drops from 0.82 to 0.8 between train and test data models for classification and from 0.71 to 0.65 for segmentation*". That is the average change from train to test data environment calculated across all configurations for ISP as well as Common Corruptions. We now added this clarification to the text, too, to avoid confusion.
> > > >
> > > > > *Fig 6.c: What's the forensics loss?*
> > > >
> > > > It is the binary cross entropy or modified Dice loss used to optimize the segmentation model. We added a clarification to the figure caption.
> > > >
> > > > > *Sec 5.2.3: Why would the data model not be part of the "shipped model"?*
> > > >
> > > > Apologies if we missed this but which part is this referring to exactly? A general note regarding this question question: under the current deployment regimes in most application domains only task models are developed and deployed.
> > > >
> > > > > *Fig 7: What does "Raw comparison" refer to? Is this the model trained on raw data? If so, it should be integrated into the the other plots so that it is easy to compare. How is the raw model trained? For segmentation, does the target segmentation mask need to be processed to fit the raw data space before demosaicing?*
> > > >
> > > > This refers to the comparison of training the task models directly on raw sensor data. While we are concerned with the setting found in most industry applications (raw access possible but ISP used, see new section 1.3), we added this comparison. To separate it from the main message we moved it to a separate subfigure. We added cross validated results and updated Figure 7 caption. We made an additional note of it in section 5.3.3 + reference to A.2 for the raw training.
> > > >
> > > > > *Fig 7: It is said this isn't possible to run similar optimisation on processed data. However, it would be possible to tune hyperparameters of gamma correction or similar. Furthermore, there is work on learning data normalisation in medical imaging [1]*
> > > >
> > > > That is true and we perform sensitivity analysis regarding the task models in relation to gamma correction (see Figure 6). Nonetheless, our framework offers the capability to comprehensively capture the entire ISP module present on acquisition hardware, which is not achievable through processed data alone.
> > > >
> > > > The paper by Michal Drozdzal and colleagues is excellent, however they are not concerned with the physics of the acqusition process for neither of the modalities (EM, CT and MRI images) they use in their paper. To that effect they also do not work on raw sensor data and do not emulate physically faithful test cases for drift validation, which is the primary focus of our work.

---

### Decision · Action_Editors · 2023-04-27

**Recommendation:** Accept as is

**Comment:**

The manuscript is improved considerably compare to the last submission and it improved even more during the discussion period. Authors have improved the writing including the discussions on the limitations of the proposed technique. Furthermore, the main missing baseline which was training on raw data has been added to the paper. After the discussion period, all reviewers recommended acceptance. Based on all above, my final recommendation is to accept the paper as is.

**Audience:**

Researchers interested in understanding and improving the robustness of ML models under distribution are the main audience of this paper.

**Claims And Evidence:**

The main claim of the paper is that differentiable data models are powerful tools for drift synthesis, drift forensics and drift adjustment. The main missing evidence from the pervious submission was comparing with training on raw data which is now added to the manuscript.